# Characterization of meiotic axis proteins in the model brown alga *Ectocarpus*

Emma I Kane[1], Lioba S Trefs [ID][2], Lena Eckert[2], Susana M Coelho [ID][1]✉ & John R Weir [ID][2]✉

## Abstract

**Most eukaryotes share core meiosis-specific genes, suggesting meiosis evolved once in the last eukaryotic common ancestor (LECA). These genes are master regulators of meiotic recombination, ensuring genetically diverse lineages. However, meiosis in organisms outside the animal, plant, and yeast lineages remains poorly understood. Core meiotic genes were recently identified in the model brown alga *Ectocarpus* but remain uncharacterised. Here, we combine bioinformatic, structural, and biochemical approaches to characterise the axial-element orthologues meiotic *Ectocarpus* HORMA-domain protein (ecHOP1) and its interactor reductional division protein 1 (ecRED1), providing insight into meiotic-recombination regulation in brown algae. We define the chromatin-binding region of ecHOP1 and show that it binds double-stranded DNA, and we find that *Ectocarpus* assembles its axial element using evolutionarily conserved principles in a unique combination. Our work lays a foundation for further studies of meiosis in brown algae and broadens understanding of the diversity and conservation of meiotic mechanisms.**

**Keywords** Meiotic recombination; HORMA; *Ectocarpus*; Brown algae; Molecular evolution
**Subject Categories** Cell Cycle; Evolution & Ecology

## Introduction

Meiotic recombination is at the center of the staggering diversity of eukaryotic life on Earth (Bolcun-Filas and Handel, 2018). Meiosis consists of two rounds of cell division, with no intervening S-phase. During meiosis I, homologous chromosomes recombine giving rise to genetically distinct haploid gametes after meiosis II (Zickler and Kleckner, 2023; Hunter, 2015). Though core meiotic genes are generally conserved across eukaryotes (Thangavel et al, 2023; Arter and Keeney, 2023; Chen and Weir, 2024), their presence and functionality can vary among species. The presence of core meiotic genes suggests meiosis likely evolved only once from the last common eukaryotic ancestor (Goodenough and Heitman, 2014;

O'Malley et al, 2019). Emerging model organisms have the potential to provide further insights into meiotic process conservation and diversity. Brown algae, members of the Stramenopile lineage, have recently emerged as powerful systems for investigating how fundamental biological processes have diversified across evolution (Coelho and Cock, 2020). Among them, *Ectocarpus* has become a key reference species, providing a well-characterized framework to study life cycle regulation and developmental complexity (Coelho, 2024) (Fig. 1A). Because brown algae evolved complex multicellularity independently of animals and plants, they offer unique perspectives on the principles and innovations that underlie multicellular organization (Denoeud et al, 2024; Coelho, 2024; Batista et al, 2024).

The life cycle involves an alternation between a haploid and diploid stage, with meiosis mediating the transition from diploid to haploid stages and syngamy reconstituting the diploid genome. Remarkably, in *Ectocarpus*, unfertilized gametes can develop parthenogenetically into haploid parthenosporophytes. These can either undergo endoreduplication to become diploid and proceed through meiosis, or, if remaining haploid, produce meiospores via apomeiosis, thereby initiating the gametophyte generation (Bothwell et al, 2010). Although the mechanisms underlying (apo)meiosis are currently unknown, this feature illustrates the plasticity of development in these fascinating yet underexplored organisms.

In organisms studied to date, prophase I chromosomes have a distinct architecture consisting of regular loops of chromatin emanating from a proteinaceous axis (Grey and de Massy, 2021; Wang et al, 2015; Kleckner, 2006; Kleckner et al, 2003). The meiotic axis is central to the control of recombination and crossover formation. The break-forming machinery assembles on the meiotic axis, whereas the breaks are made in the chromatin loops catalyzed by the topoisomerase-like Spo11 (Blat et al, 2002). In yeasts and mammals, the HORMA domain protein Hop1/HORMAD1 facilitates the connection between the double-strand DNA break (DSB) machinery and the axis (Stanzione et al, 2016; Dereli et al, 2024; Rousova et al, 2021). Hop1 is loaded onto the axis in a manner partially dependent on the AAA+ ATPase Pch2 (TRIP13 in mammals) (Raina and Vader, 2020), where it binds to the axial component Red1 (SYCP2) (West et al, 2019). It is thought, though not formally proven, that Red1/SYCP2 is itself recruited to chromosomes via an interaction with cohesin complexes, due to both co-localization on chromosomes, proteomics experiments,

[1]Department of Algal Evolution and Development, Max Planck Institute for Biology, Tübingen, Germany. [2]Friedrich Miescher Laboratory of the Max Planck Society, Tübingen, Germany. ✉E-mail: susana.coelho@tuebingen.mpg.de; john.weir@tuebingen.mpg.de

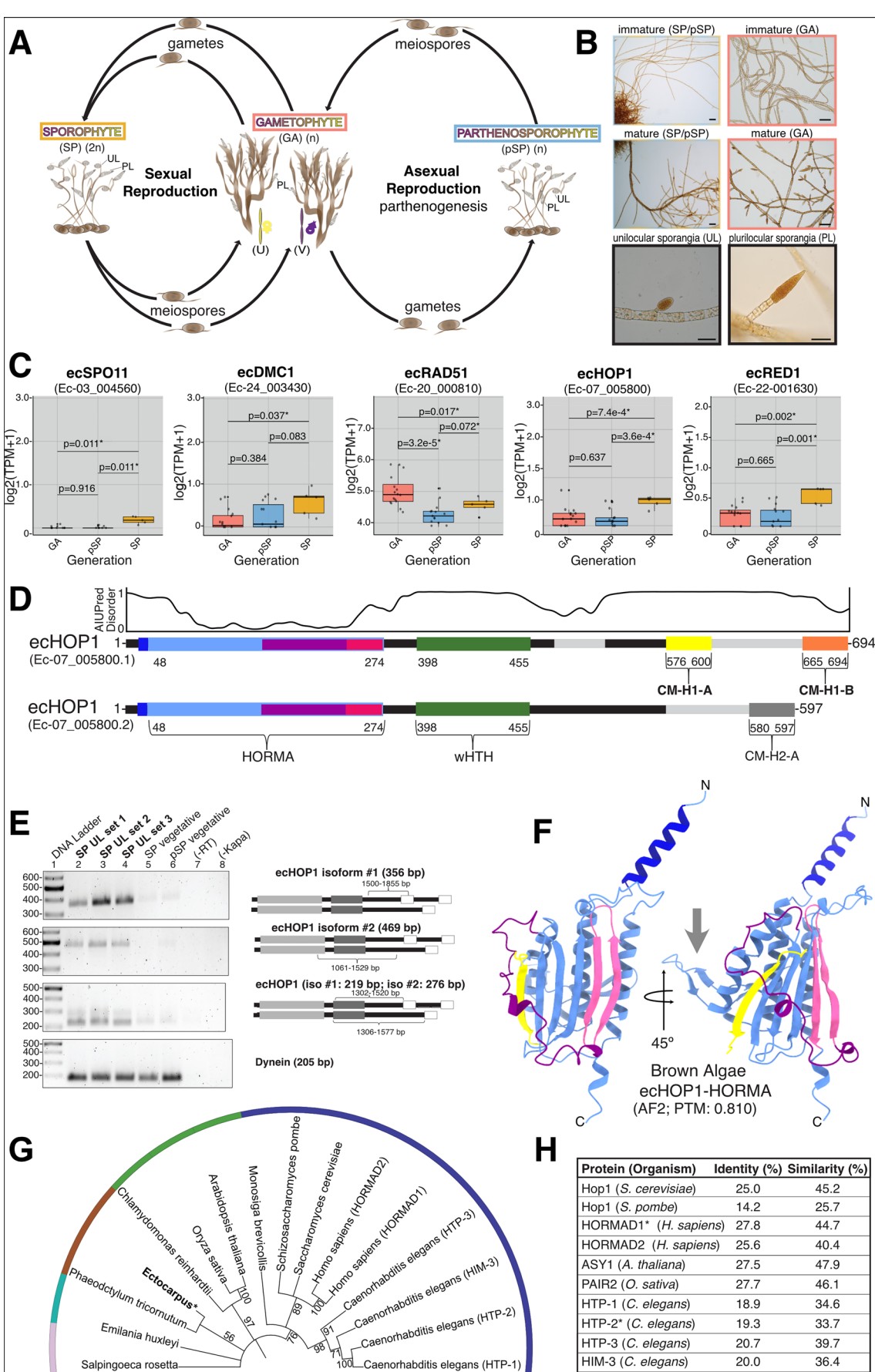

**Figure 1. Identification of core meiotic genes across the unique life cycle of *Ectocarpus*.**

(A) Overview of the *Ectocarpus* life cycle, which alternates between haploid (*n*) and diploid (2*n*) stages. Male and female gametes fuse to form a zygote, which develops into the sporophyte (SP; yellow). After meiosis in the unilocular sporangia (UL), the release of meiospores from the sporophyte leads to the formation of gametophytes (GA; salmon), completing the sexual cycle. In the absence of male gametes, female gametes can undergo parthenogenesis, producing clones that develop into parthenosporophytes (pSP; cyan), which are morphologically indistinguishable from diploid sporophytes. These pSP produce (*n*) meiospores via a so far unknown process of (apo)meiosis (Bothwell et al, (2010)). (B) Microscopy images of immature and mature GA, SP, and pSP, alongside images of unilocular sporangia (UL) and plurilocular sporangia (PL). The SP that successfully complete the sexual life cycle to form GA are highlighted, with a focus on the PL and UL (where meiosis occurs). Scale bars = 40 microns. (C) Boxplot showing the expression of core meiotic and HORMA domain-containing genes, based on published RNAseq data. Data are presented for GA, pSP, and SP samples from male (Ec32) and female (Ec25) GA/pSP, as well as SP (Ec17) strains. Colors correspond to those in (A, B), visualizing gene expression at different life cycle stages. Statistical significance between samples is indicated (*p* < 0.05), and boxplots were generated in R. Box plots show the median (center line), the IQR (box: 25th–75th percentiles), and the whiskers indicate 1.5× the IQR. Outliers are shown as individual dots. This approach was chosen to preserve consistent visual scaling across genes with very different expression levels, such as low-expression genes (i.e., *ecHOP1* and *ecRED1*) versus highly expressed genes (i.e., *ecRAD51*), and to allow sufficient vertical space for statistical annotations. Each gene includes *n* = 9–12 biological replicates for GA and pSP (from Ec25 and Ec32), and *n* = 3 replicates for SP (from Ec17). Data are taken from publicly available sources (Lipinska et al, 2017; Data ref: Lipinska et al, 2017; Lotharukpong et al, 2024; Data ref: Lotharukpong et al, 2024). (D) Domain architecture of ecHOP1 isoforms generated utilizing the sequences in ORCAE (Ec-07_005800.1 for isoform #1 and Ec-07_005800.2 for isoform #2), coupled with AlphaFold2 modeling. The boundaries for the putative CMs were determined from AlphaFold 2 models and alignment with previously identified CMs. Gray regions denote the variable sequences between the isoforms. Figure EV1A shows the details of the differences between the isoforms. Disorder propensity based on AIUPred (Erdős and Dosztányi, 2024) is plotted above for isoform #1. (E) RT-PCR identified the presence of both isoforms from replicate uniloc samples collected from SP tissue (bold), and vegetative (non-reproductive) tissue from SP and pSP. Primer design was used to ensure the specificity of amplified regions that are unique between both isoforms, as well as overlapping regions to identify both isoforms in the same sample (highlighted in the graphics below each result). RT-PCR controls as well as primers amplifying the Dynein gene were utilized. (F) AlphaFold2 model of the ecHOP1 HORMA domain (light blue) bound in cis to a predicted CM (yellow). The *N*-helix is shown in dark blue, the safety belt in purple, and the buckle in pink. The β3″ hairpin is denoted with a gray arrow. (G) Maximum-likelihood phylogenetic tree of meiotic HORMA domain-containing proteins from major eukaryotic lineages. Full-length protein sequences from known and putative orthologs were analyzed using IQ-TREE2 with the LG + G4 model and 1000 ultrafast bootstrap replicates. Colors indicate eukaryotic supergroups: light pink—Cryptista, cyan—Haptista, brown—SAR, green —Archaeplastida, dark blue—Opisthokonta. The SAR group includes *Ectocarpus siliculosus* (ecHOP1) and *Phaeodactylum tricornutum* (PtHOP1); Opisthokonta includes *Homo sapiens* (HORMAD1/2), *Saccharomyces cerevisiae* (HOP1), *Schizosaccharomyces pombe* (HOP1), *Caenorhabditis elegans* (HTP-1/2/3, HIM-3), and *Monosiga brevicollis* (putative HOP1); Archaeplastida includes *Arabidopsis thaliana* (ASY1), *Oryza sativa* (PAIR2), and *Chlamydomonas reinhardtii* (CrHOP1); Haptista and Cryptista are represented by *Emiliania huxleyi* (EhHOP1) and *Salpingoeca rosetta* (SrHOP1), respectively. ecHOP1 is highlighted to show its phylogenetic relationship to canonical meiotic HORMAs. Bootstrap support values are shown at nodes. (H) Pairwise alignment of the ecHOP1 HORMA domain (residues 1–274) with meiotic HORMA domains from other major model organisms. Alignments were performed using EMBOSS Needle (EMBL-EBI) to evaluate sequence similarity between ecHOP1 and selected HORMA proteins from diverse eukaryotic lineages. Source data are available online for this figure.

and the presence of cohesin binding motifs in Red1/SYCP2 (Sun et al, 2015; Köhler et al, 2017; Xu et al, 2019; Li et al, 2020). Hop1 has been shown to be phosphorylated in response to DSB formation by the ATM/ATR kinases (Carballo et al, 2008; Penedos et al, 2015). Phospho-Hop1 activates the meiotic checkpoint, which in yeast is mediated by the kinase Mek1, and controls progression through meiosis (Wu et al, 2010; Chen et al, 2018; Xu et al, 1997).

Pairing between homologous chromosomes manifests itself in the form of the synaptonemal complex (SC), which forms along the length of the meiotic axis, possibly through a direct interaction between Zip1/SYCP1 and Red1/SYCP2 (Adams and Davies, 2023). As chromosomes synapse, the meiotic axis is remodeled, suppressing further DSB formation (Wojtasz et al, 2009). The length of the meiotic axis, relative to the size of the linear chromosome, has been shown to control the number of crossovers. Chromosomes with shorter axes have fewer crossovers, with longer axes having more (Ruiz-Herrera et al, 2017). Therefore, understanding the principles of meiotic axis assembly is key to understanding the initiation and regulation of meiotic recombination.

HORMA domain proteins play a critical role in DNA repair and checkpoint signaling (Gu et al, 2022). HORMA domains undergo conformational transformations from an "open" to "closed" state catalyzed through interactions with closure motifs (CMs). Therefore, one HORMA domain can have distinct interactomes dependent on its topological state. Meiotic HORMA domain proteins (Hop1 in yeast, HORMAD1 in mammals, Asy1 in plants, HTP-1,2,3 and HIM3 in nematodes) (Prince and Martinez-Perez, 2022) have all so far been found to contain an *N*-terminal HORMA domain and one *C*-terminal CM, with the exception of HTP-3 that contains six CMs (Kim et al, 2014). Due to the presence of these *cis*

CMs, it is expected that meiotic HORMA domains have a self-bound "ground-state." Meiotic HORMA proteins can associate with the axis through a CM in other axial proteins (Red1 in yeast, SYCP2 in mammals, Asy3 in plants). Furthermore, a second recruitment pathway was recently shown for budding yeast Hop1, which can also associate directly with nucleosomes via a chromatin binding region (Milano et al, 2024; Heldrich et al, 2022). The chromatin-binding region of Hop1 has been shown to be highly variable throughout evolution, including complete absence in mammals and nematodes (Milano et al, 2024).

To better understand the conservation and diversity of meiotic protein function across eukaryotes, we investigated the role of meiotic proteins in *Ectocarpus*. *Ectocarpus* has been shown to have both a Spo11 ortholog (Brinkmeier et al, 2022), and brown algae possess a putative SC (Toth and Markey, 1973). Moreover, candidate genes for both Red1 and Hop1 have been identified across a range of eukaryotes, including *Ectocarpus* (Tromer et al, 2021). Through genomic and structural analysis, we confirm the identity of these orthologs and biochemically characterize the core axial protein ecRED1 and its associated regulatory protein ecHOP1. We then characterize the DNA-binding domain of ecHOP1, which binds preferentially to dsDNA, analogously to the nucleosome-binding domain of budding yeast Hop1. Our findings reveal species-specific adaptations, including novel transcriptional isoforms and the presence of multiple CMs in ecHOP1. Together, our findings suggest that the meiotic axis in brown algae exhibits a blend of features seen in the meiotic axes of other eukaryotes, with functional plasticity arising from the differential utilization of a conserved set of meiotic genes across the tree of life.

# Results and discussion

## Life cycle-dependent expression of axial element orthologs

The life cycle of *Ectocarpus* involves an alternation between haploid male and female gametophytes (GAs) and diploid sporophytes (SP). Meiosis occurs in the unilocular (UL) reproductive structures in the sporophyte generation. *Ectocarpus* may also undergo an alternative asexual cycle via parthenogenesis, where female gametes develop autonomously (without fusion with a male gamete) into haploid parthenosporophytes (pSP). These pSP are morphologically indistinguishable from diploid sporophytes and at maturity undergo (apo)meiosis to produce haploid spores (meiospores). Therefore, in addition to a classical reductional meiosis, these organisms may undergo an alternative form of non-reductive meiosis (Bothwell et al, 2010) (Fig. 1A,B).

We used published RNAseq data to investigate the transcript abundance of genes related to meiosis during the complex life cycle of *Ectocarpus* (Fig. 1C). We focused on orthologs involved in recombination and SC formation. We found that ecSPO11, the conserved initiating factor for meiotic recombination (Bergerat et al, 1997; Keeney et al, 1997), was specifically upregulated in (diploid) sporophytes, where reductional meiosis occurs. Similarly, the meiotic-specific recombinase ecDMC1 (Bishop et al, 1992) showed a specific upregulation in diploid sporophytes. In contrast, the recombinase ecRAD51, which is involved in both meiotic and somatic DNA repair, did not show an expression pattern specific to meiotic tissues.

Previous bioinformatics work provided candidate genes for the *Ectocarpus* axial protein RED1 (Tromer et al, 2021) (from hereon ecRED1) and its associated HORMA domain-containing partner HOP1 (ecHOP1). We therefore also investigated the transcript abundance of these genes. In line with the expression patterns for meiosis-specific ecSPO11 and ecDMC1, ecRED1 and ecHOP1 both showed specific expression in sporophyte tissue (Fig. 1C). Together, our data demonstrates that core meiotic genes are upregulated in stages of the life cycle where meiosis occurs, consistent with a role of these genes specifically in reductional meiosis but not in (apo) meiosis in the parthenosporophyte. It is currently elusive what mechanisms underlie apomeiosis, but given the lack of expression of ecHOP1/ecRED1 in pSP, it is possible that these organisms either use other axis-like factors or bypass the need for a classical axis during this process.

## Identification and computational analyses of ecHOP1

ecHOP1 has two transcriptional isoforms reported through ORCAE (online resource for community annotation of eukaryotes) (Ec_07-005800.1 and Ec_07-005800.1, respectively) (Sterck et al, 2012) (Fig. 1D). Both isoforms of ecHOP1 have a shared *N*-terminal region of the protein, while encoding a divergent *C*-terminal region (Figs. 1E and EV1A). We utilized isoform-specific RT-PCR primers with RNA extracts from *Ectocarpus*, specifically unilocs collected during the SP life stage and vegetative tissue (non-reproductive) from both SP and pSP life stages. In these experiments, we found direct evidence of expression of ecHOP1 isoforms 1 and 2 (Fig. 1E).

In the absence of experimental structural information on ecHOP1, we utilized AlphaFold2 (Jumper et al, 2021) to gain insights into the structure and function of the ecHOP1 isoforms (Figs. 1F and EV1B). Both isoforms are predicted to contain a conserved *N*-terminal HORMA domain, a winged helix–turn–helix (wHTH) domain, which may bind to chromatin, as in budding yeast Hop1 (Milano et al, 2024) (Fig. 1D, green). To place ecHOP1 in an evolutionary context, we constructed a maximum-likelihood phylogenetic tree using known and putative meiotic HORMA domain-containing proteins from major eukaryotic lineages (Fig. 1G). This analysis confirms that ecHOP1 clusters within the SAR lineage and is evolutionarily related to canonical meiotic HORMADs. Structural comparison of the predicted HORMA domain of ecHOP1 (aa. 50–274) with those from nematode (PDB 4TZM (Kim et al, 2014)) and mammalian (PDB 8J69 (Wang et al, 2023)) orthologs revealed high structural similarity, with C-alpha RMSDs of 1.142 Å and 0.927 Å, respectively (Fig. EV1C). Additionally, ecHOP1 also has a high level of sequence similarity with other well-studied meiotic HORMADs (Fig. 1H).

Close inspection of the AF2 structure of the ecHOP1 HORMA domain revealed two features that have not previously been described in other meiotic HORMA domains. Firstly, we observe an extended α-helix in the *N*-terminal region, which we term *N*-helix (Fig. 1F, Fig. EV1B dark blue). We also find a β-hairpin between residues 94 and 103, inserted between the β3 and αB regions (Fig. 1G gray arrow, Fig. EV1B, green box). We term this region the β3″ hairpin, which is conserved within brown algae (Fig. EV1D). Our findings confirm the core domains of ecHOP1, validate both isoforms, highlight the conserved HORMA domain's similarity to well-studied counterparts, and uncover a novel feature: multiple CMs in one isoform.

## Interaction of ecHOP1-HORMA with ecHOP1 CMs

To investigate ecHOP1 further, we turned to recombinant protein approaches. Given our observation that ecHOP1 also contains an *N*-terminal extended α-helix (*N*-helix, Fig. 1F, Fig. EV1B in dark blue), we utilized a construct with the full *N*-terminal region consisting of residues 1–274, which we call ecHOP1-HORMA, even though the HORMA domain only starts at amino acid 48. We expressed ecHOP1-HORMA in *E. coli* with an *N*-terminal 2×Strep-II and purified the protein to homogeneity (Figs. 2A and EV2A). Size-exclusion chromatography coupled with multi-angle light scattering (SEC-MALS) revealed a species of ecHOP1-HORMA at 34 kDa ($+/-7.5\%$) (Fig. 2B), consistent with a monomeric ecHOP1-HORMA (theoretical Mw of 36.19 kDa).

Since previously studied meiotic HORMADs interact with their own CMs—located in the extreme *C*-termini of meiotic HORMADs —in *cis*, we asked whether AlphaFold2 could predict potential CMs (or simple motifs; M) in both of the ecHOP1 isoforms. Curiously, AlphaFold2 suggested two potential motifs in isoform 1 (CM-H1-A and CM-H1-B) comprising residues 567–600 and 665–694, respectively, and isoform 2 (M-H2-A) in the very *C*-terminal region (residues 580–597) (Fig. 1D, yellow, orange and gray boxes, Fig. 2C–E). These motifs contain conserved arginine residues, potentially important for HORMA domain interaction as previously reported in other CMs (West et al, 2017; Kim et al, 2014) (Fig. 2F), but are not otherwise very similar to one another. This

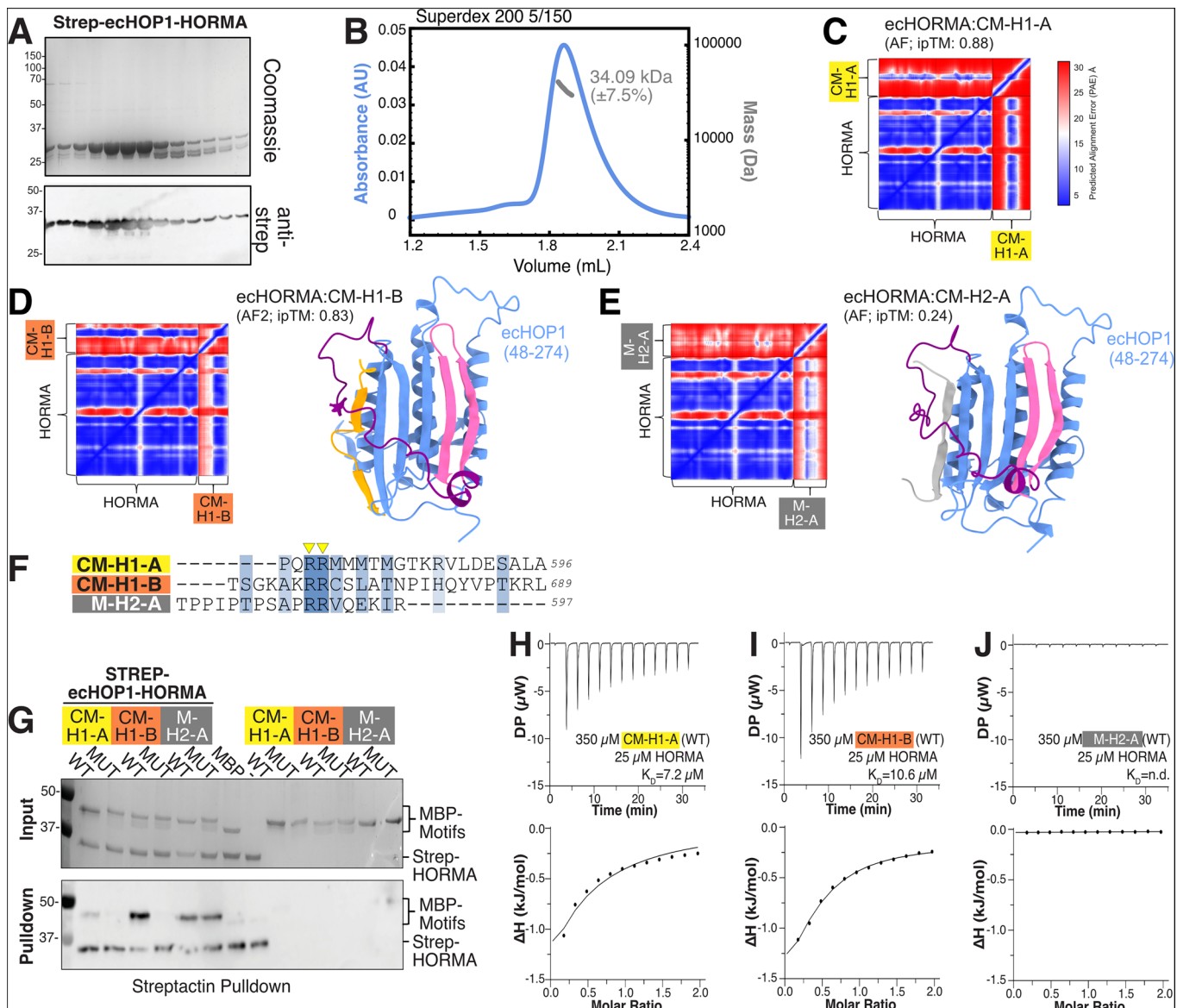

**Figure 2.  In vitro characterization of the self-closing HORMA conformational switches of ecHOP1 with all putative motifs.**

(A) Upper panel—Coomassie-stained SDS-PAGE gel of a final size exclusion purification of ecHOP1-HORMA. Lower panel—anti-Strep-II western blot of the same fractions. The corresponding chromatogram can be found in Fig. EV2, where the Coomassie-stained gel and western blot is also shown again. (B) SEC-MALS of ecHOP1-HORMA run on a Superdex 200 5/150 column. The main peak corresponds to a measured MW of ~34 kDa. (C) Predicted alignment error (PAE) plot of AlphaFold2 prediction of ecHOP1-HORMA with CM-H1-A (as shown in Fig. 1F). (D) AlphaFold2 model of ecHOP1-HORMA plus CM-H1-B with corresponding PAE plot. (E) AlphaFold2 model of ecHOP1-HORMA plus CM-H2-B with corresponding PAE plot. (F) Aligned ecHOP1 CMs with the conserved arginine residues highlighted with yellow triangles. (G) Pull-down assays with the WT and mutant (R/A) CMs of ecHOP1. 2xStrepII-ecHOP1-HORMA was used as bait for the prey MBP-tagged CMs. In addition, MBP-CMs were used alone as a control for background binding to the Streptactin beads. (H–J) Isothermal titration calorimetry (ITC). 350 µM of indicated MBP-tagged CMs was titrated against 25 µM of ecHOP1-HORMA in the cell. Buffer–buffer controls were run and subtracted from the experimental data to yield the heats shown. Binding curves were fitted in the software, and the determined $K_D$ is shown. Source data are available online for this figure.

sequence degeneracy has previously been observed for other meiotic CMs (West et al, 2017; Kim et al, 2014). Analysis of the potential interactions for CM-H1-A and CM-H1-B seems to show that the side chain chemistry of the interactions is favorable, and that these residues are also conserved in brown algae (Fig. EV3). Close analysis of CM-H1-A reveals that one of the two conserved arginines (R583) is predicted to make side-chain H-bonds with E227 and also the main chain of W220, both in the safety belt. In

CM-H1-A, the arginines are also followed by three methionine residues. In the model, these make hydrophobic contacts with the side chain carbons of R226 and K208, and with W210 and A208. These residues are shared between the safety belt and the core of the HORMA domain (Fig. EV4). A similar pattern of interactions was predicted for CM-H1-B (Fig. EV3D).

To test potential interaction between ecHOP1-HORMA and the three predicted CMs, we produced recombinant ecHOP1 CMs fused to

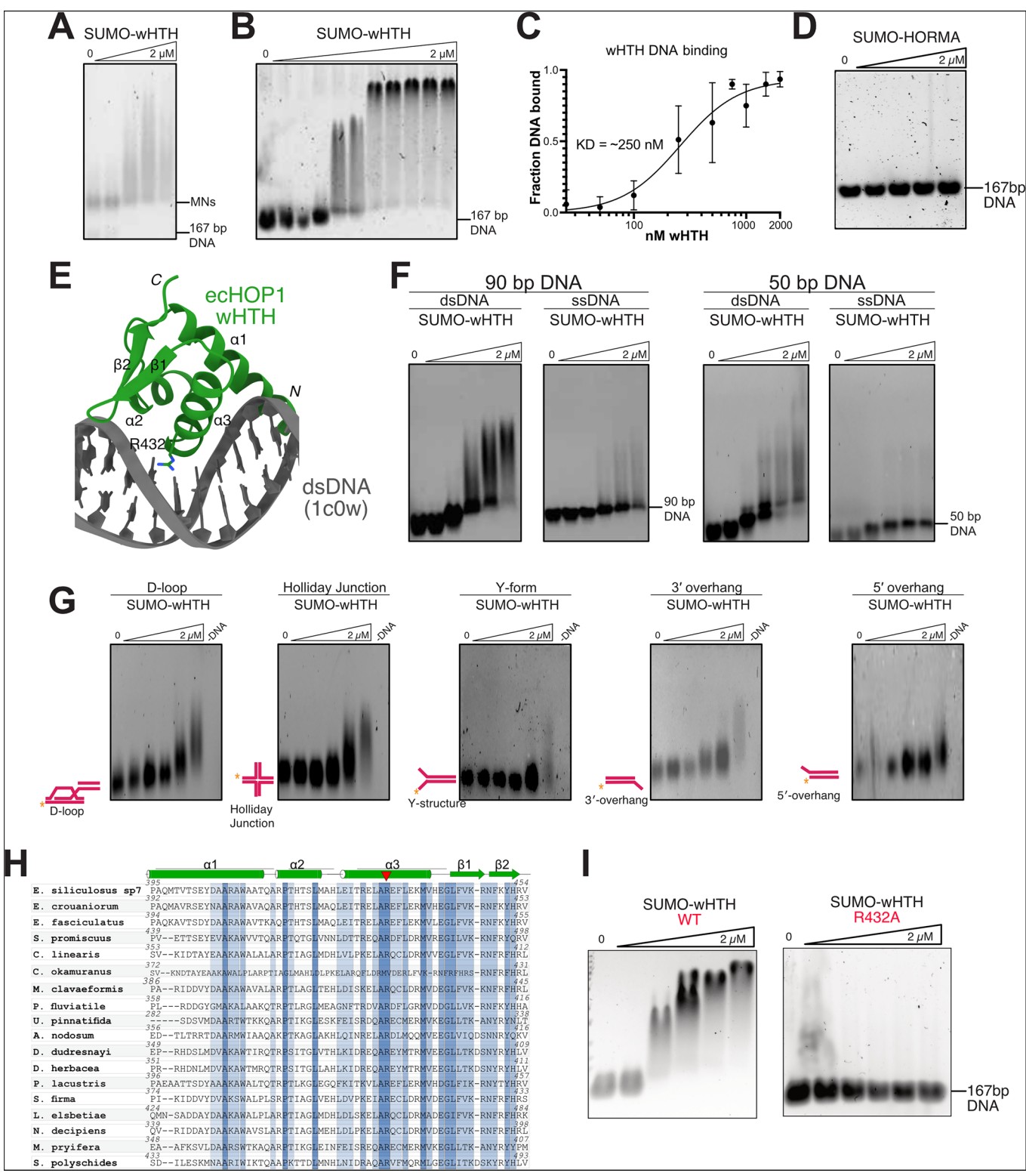

an *N*-terminal maltose-binding protein (MBP) to improve solubility. For each motif, we produced a mutant variant where the commonly conserved arginine residues were mutated to alanine (Fig. 2, yellow triangles). We first tested the interaction between Strep-II tagged ecHOP1-HORMA and MBP-CMs in a pull-down on Strep Tactin-XT beads (Fig. 2G). Here, we observed an interaction for all three CMs. For the ecHOP1 isoform 1 CMs CM-H1-A and CM-H1-B, the binding was also disrupted by the introduced mutations, which was not the case for the isoform 2 motif (M-H2-A), where both the wild-type and mutant sequences appeared to bind to ecHOP1-HORMA.

Figure 3. wHTH domain of ecHOP1 binds dsDNA.

(A) EMSA of SUMO-tagged ecHOP1-wHTH (hereon SUMO-wHTH) titrated against 50 nM of nucleosomes wrapped in 167 bp Widom DNA. (B) EMSA of SUMO-wHTH titrated against 50 nM of 167 bp DNA. (C) Binding curve based on triplicate experiments as shown in (B). Curve fitting carried out in Prism required the use of the Hill Coefficient for the best fit ($h = 1.7$). Error bars show the SD from three replicates. (D) ecHOP1-HORMA domain was also titrated against 167 bp Widom DNA in an EMSA experiment as a negative control. (E) Composite structural model. The AlphaFold2 prediction of ecHOP1-wHTH (residues 398–454) superimposed onto the wHTH in DtxR (PDB:1C0W). The dsDNA from 1C0W is shown (gray) and the model should approximate the DNA-binding mode of SUMO-wHTH. R432 of ecHOP1 sits in the middle of the candidate DNA binding helix, and is proximal to the sugar-phosphate backbone of the dsDNA. (F) EMSAs of SUMO-wHTH binding to 90-mer dsDNA and ssDNA. (G) EMSAs showing binding of ecHOP1-wHTH to different DNA structures as indicated. (H) MSA of multiple wHTH domains from across Stramenopiles in addition to A. thaliana, C. reinhardtii, and P. falciparum sequences. The location of R432 of ecHOP1 is indicated by the red arrow in the middle of the α3-helix. (I) EMSA showing the titration of SUMO-wHTH$^{R432A}$ mutant against 167 bp Widom DNA. Replicates (including these EMSAs) are shown in Fig. EV6D,E. Source data are available online for this figure.

To gain further insights into CM to ecHOP1–HORMA interactions, we determined the relative binding affinities by isothermal titration calorimetry (ITC) (Fig. 2H–J). Under these conditions, we could further confirm the interaction of CM-H1-A and CM-H1-B with ecHOP1$^{HORMA}$. The measured dissociation constants ($K_D$) for CM-H1-A and CM-H1-B were 7.2 and 10.6 μM, respectively. We did not detect an interaction for M-H2-A (Fig. 2J). Based on this, and that the mutations in M-H2-A did not disrupt binding in the pull-down experiment, we conclude that M-H2-A is not a functional CM (hence our referral to this as "M"-H2-A, rather than "CM"), and that ecHOP1 isoform 2 does not contain a CM for cis-binding.

One possible implication of this arrangement would be that ecHOP1 isoform 1 can bind to two ecHOP1-HORMA domains simultaneously via CM-H1-A and CM-H1-B. To test whether there is sufficient distance between CM-H1-A and CM-H1-B, we utilized AlphaFold2. We modeled the C-terminus (residues 212–694) of ecHOP1 isoform 1 together with two copies of full-length ecHOP1 isoform 2 (which does not contain a functional CM). This model shows that simultaneous binding of two ecHOP1-HORMA domains to the C-terminus is theoretically possible (Fig. EV5).

## DNA-binding capabilities of ecHOP1

As described above, AlphaFold modeling of ecHOP1 predicted the presence of a wHTH domain (residues 398–455, Fig. 1E green). This domain is conserved among HOP1 SAR, and Haptista clades (Fig. EV6A), though occasionally as a twin wHTH domain (Milano et al, 2024). An expanded version of the wHTH domain in budding yeast (which also contains a PHD and a second wHTH domain) was recently found to bind to nucleosomes (Milano et al, 2024). Two species from our subset, P. terocystis and P. falciparum showed twin wHTH domains, which have also been shown to occur elsewhere in evolution (Milano et al, 2024). To investigate the chromatin-binding properties of the ecHOP1-wHTH domain, we produced recombinant SUMO-tagged ecHOP1-wHTH, purified this to homogeneity (Fig. EV6B), and evaluated DNA binding via electrophoretic mobility shift assays (EMSAs).

We first asked whether ecHOP1-wHTH might be able to bind to mononucleosomes wrapped in 167 bp Widom DNA (Lowary and Widom, 1998), as has been shown for the equivalent region of S. cerevisiae Hop1 (Milano et al, 2024). However, we only observed weak binding under these conditions (Fig. 3A). We then tested whether wHTH could bind to the 167 bp Widom dsDNA alone and observed robust binding, for which we determined an apparent $K_D \sim 250$ nM (Fig. 3B,C). As a comparison, we also measured DNA

binding for ecHOP1-HORMA, which did not show any clear DNA binding (Fig. 3D).

We carried out a DALI (Holm, 2020) search with the AlphaFold2 model of ecHOP1-wHTH to find the most similar structures of wHTH domains in complex with nucleic acids in the PDB. We focused on the structure of DtxR with dsDNA (PDB ID 1C0W (Pohl et al, 1999)). We superimposed the AF2 model of ecHOP1-wHTH onto the iron-dependent repressor DtxR bound to dsDNA, which gave a C-alpha RMSD of 0.727 Å (Fig. 3E). In this structure, the wHTH binds to dsDNA via the major groove, with α3 of wHTH sitting in the major groove. If ecHOP1-wHTH is binding via a similar mechanism, we reasoned that this would strongly prefer dsDNA over single-stranded DNA (ssDNA). We evaluated this in an EMSA using both 90-mer and 50-mer ssDNA and dsDNA and found that ecHOP1-wHTH indeed binds more tightly to dsDNA (Fig. 3F). We next asked whether ecHOP1-wHTH might have a preference for particular DNA shapes, which could either represent an axis formation preference on certain DNA secondary structures or a role in binding to certain recombination intermediates. We utilized fluorescently labeled DNA substrates (Fig. 3G, fluorophore position indicated by orange asterisk) in an EMSA experiment. While the ecHOP1-wHTH domain showed comparatively weak binding to a Y-structure and a 5′ overhang, we conclude that ecHOP1-wHTH has no strong DNA structure preference (Fig. 3G).

We produced a MSA of HOP1-wHTH from Stramenopiles (Fig. 3H), and expanded this further within the SAR and Haptista clades (Fig. EV6A). By combining the AF2 model superimposed onto dsDNA and the MSA, we identified residues within α3 of the ecHOP1-wHTH domain that could mediate DNA binding. R432 is positioned within the α3 helix and is proximal to the sugar-phosphate backbone of the DNA. We produced mutant recombinant ecHOP1-wHTH with an R432A mutation. This mutant behaved well during purification and showed a similar profile in a circular dichroism (CD) experiment (Fig. EV6C). We detected no dsDNA binding with ecHOP1-wHTH$^{R432A}$ when compared with the wild-type counterpart (Figs. 3I and EV6D,E). These results establish that ecHOP1-wHTH binds preferentially and robustly to dsDNA, likely via interactions mediated by α3, with R432 playing a critical role in this binding, as demonstrated by the loss of DNA interaction in the R432A mutant.

## Characterization of the ecHOP1 interactor ecRED1

Next, we expanded our studies to include the predicted Red1 ortholog in Ectocarpus (from here on ecRED1). Analysis of the AlphaFold2 predicted structure of ecRED1 shows that the protein contains an

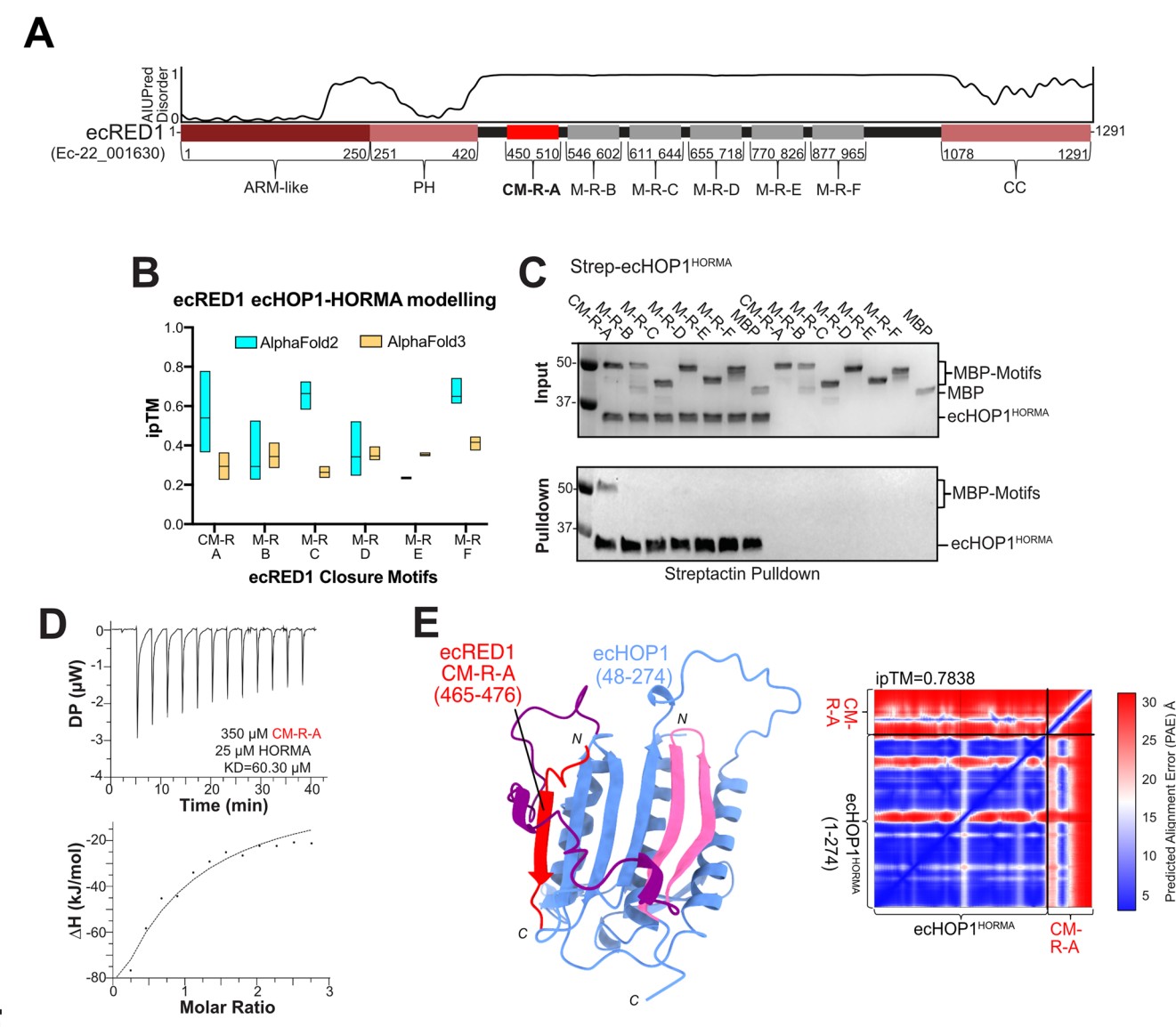

**F**

**HORMA to candidate closure motif side chain interaction table from top scoring models**

| HORMA | CM-H1-A | CM-H1-B | M-H2-A | CM-R-A | M-R-B | M-R-C | M-R-D | M-R-E | M-R-F |
|---|---|---|---|---|---|---|---|---|---|
| W220 | R583 | R672 | | | | | R662 | | R938 |
| N213 | R583 | R672 | | R469 | R551 | | S660 | T800 | R938 |
| A228 | M584 | C673 | | | | | | | A939 |
| L209 | M585 | S674 | | A570 | T553 | A629 | G664/E692 | V803 | H940 |
| R226 | M586 | L675 | Q561 | V471 | | T630 | S685 | | V941 |
| I207 | T587 | A676 | V562 | C472 | G574/A575 | A631 | T666 | A806 | Q942 |
| L205 | T590 | I680 | | L474 | V577 | R633 | W668 | A808 | |
| R203 | R592 | Q682 | | | | A635/T637 | I670 | G810 | |
| R201 | L594 | V684 | | | | T637 | E672 | | |

*N*-terminal armadillo (ARM) domain, a mid-region Pleckstrin homology (PH) domain, and a *C*-terminal coiled-coil (CC) domain (Figs. 4A and EV6A–D), a domain arrangement widely found conserved in Red1 orthologs (Feng et al, 2017; West et al, 2019). The CC domain of Red1 (yeast) has been shown to form homotetramers (West et al, 2019), whereas other orthologous proteins may form homotetramers (SYCP3 (Syrjänen et al, 2014)) or heterotetramers (SYCP2-SYCP3 (West et al, 2019)). AlphaFold2 modeling of the ecRED1 CC region (residues 1075–1291) suggests that the formation of homotetramers would be possible for ecRED1 (Fig. EV7E).

◀ **Figure 4.   Characterization of the putative interactor of ecHOP1 (ecRED1).**

(A) Domain architecture of ecRED1, with the sequence from ORCAE (Ec-22_001630), coupled with domain boundaries determined via AlphaFold2. The ARM-like, PH, and CC domains were modeled individually in AlphaFold2 (Fig. EV7). Determination of multiple CMs in ecRED1 was made via MSA and AlphaFold2 modeling, where multiple models were generated with truncations of ecRED1. (B) Summary of ipTM values (interchain confidence score) based on the top 5 scores from 25 predictions for both AlphaFold2.3.2 and AlphaFold3 (run on www.alphafoldserver.com). The line in the bar plot shows the median value for the predictions. (C) Pull-down assays with 2×StrepII-ecHOP1-HORMA domain and the putative MBP-tagged CMs on Streptactin beads. (D) 350 μM of MBP-CM-R-A was titrated against 25 μM ecHOP1-HORMA in the cell. A buffer–buffer control experiment was subtracted from the data to provide the heats shown. A $K_D$ of 60.3 μM was determined. (E) AlphaFold2 model of ecHOP1-HORMA in complex with ecRED1-CM-R-A (red). The predicted alignment error plot for the model is also shown. (F) Table of predicted side-chain interactions (see Fig. EV3 for a reference) for the candidate closure motifs and the ecHOP1 HORMA domain. Source data are available online for this figure.

In other species, Red1 orthologs have been shown to contain a CM that binds to the HORMA domain of Hop1 ortholog (West et al, 2019; Woltering et al, 2000). To find potential CMs in ecRED1, we carried out AlphaFold2-multimer (Evans et al, 2021) predictions of ecHOP1-ecRED1 sequences. The algorithm predicted six potential CMs (CM-R-A to M-R-B to M-R-F), with varying levels of confidence based on the interchain confidence score (ipTM) (Fig. 4B; for models and PAE plots, see Fig. EV8). We also ran our candidate motifs using AlphaFold3 (Abramson et al, 2024). Here the algorithm gave overall lower ipTM scores, and with no differences between the motifs (Fig. 4B). We therefore undertook a biochemical approach to validate the putative ecRED1 CMs. Curiously, the number of predicted ecRED1 CMs, six, is also the same number of CMs found in *C. elegans* HTP-3 (Kim et al, 2014).

We expressed and purified all predicted ecRED1 candidate CMs as *N*-terminal MBP-fusion proteins. In pull-downs and ITC on StrepII-tagged ecHOP1-HORMA, only CM-R-A demonstrated binding to the ecHOP1 HORMA domain (Fig. 4C,D). We then performed a phylogenetic analysis of the ecHOP1 and EcRED1 candidate CMs across the brown algal family, which revealed that motifs such as "(K/R)MMMT(K/R)" in CM-H1-A and "(K/R)C(S/G)LA(K/R)" in both CM-H1-B and CM-R-A are conserved (Fig. EV9). Notably, this conservation of the C(S/G)LA motif across both a self-containing CM (CM-H1-B) and an interactor-derived CM (CM-R-A) highlights a conserved feature of brown algal meiotic HORMA domain regulation.

ITC results showed that CM-R-A exhibited a binding affinity of 60 μM (Fig. 4D); this is nearly 10× weaker than CM-H1-A of ecHOP1 (Fig. 3H). Curiously, this is inverse to observations from yeast, where Red1 CMs exhibit stronger binding to HORMA domains than Hop1 CMs (West et al, 2017). The remaining putative ecRED1 candidate motifs could not be validated through in vitro analyses. Based on the predicted binding mechanism for the experimentally validated CM-H1-A and CM-H1-B, we examined the predictions for CM-R-A to M-R-F and looked for similarities. Unlike the clearly non-functional M-H2-A, all of the ecRED1 CMs showed extensive plausible interactions between the CM and both the safety belt and core of ecHOP1 (Fig. 4F). The conservation of the ecRED1-CMs within the brown algal family renders the possibility that under specific biological conditions they may either act as CMs or have another important function (Fig. EV9).

It has been previously shown that yeast Red1 is highly phosphorylated, especially in response to meiotic DSB formation (Kar et al, 2022; Koch et al, 2024). Also in yeast, the phosphorylation of Hop1 on threonine 318 (pT318) is required for the recruitment and activation of the kinase Mek1 (Penedos et al, 2015). Both proteins are also SUMOylated in yeast (Bhagwat et al,

2021; Eichinger and Jentsch, 2010). We therefore utilized predictive tools to establish whether this might be the case for ecHOP1 and ecRED1. Both proteins showed several low-confidence SUMOylation sites and one putative SUMO interaction site (SIM) (Fig. EV10). AlphaFold modeling of ecHOP1 or ecRED1 with both human SUMO and the putative *Ectocarpus* SUMO sequence (Ec-27_001840) also showed some potential interaction sites between ecRED1 or ecHOP1 and SUMO, but these did not overlap with the predicted SIMs. Regarding phosphorylation, it is not known whether the kinases in *Ectocarpus* have the same consensus sites as in other model organisms. Therefore, we asked whether the serines and threonines could be phosphorylated based on an unbiased accessibility prediction. Several sites are, including serine and threonine residues included within, or peripheral to, the experimentally non-functional ecRED1-CMs.

Meiosis is a ubiquitous feature of eukaryotic life, yet even among conventional model organisms, there is considerable diversification of the molecular machinery underpinning meiosis (Arter and Keeney, 2023). Utilizing tissue-specific transcriptomic data, we have confirmed the meiotic expression of two chromosomal-associated proteins, ecHOP1 and ecRED1. Structural modeling combined with biochemical and biophysical assays suggests that these proteins share similarities with orthologs in other eukaryotic organisms, but also reveal unique features.

Given that, in other species, Hop1 is a regulatory factor for the formation of meiotic DSBs, we wanted to understand how ecHOP1 might be localized to chromosomes. We find two possible pathways. One is via a direct interaction between the CMs in ecRED1, a second is via the DNA-binding properties of the wHTH domain (Fig. 3, Fig. 5 green octagon). Unlike in budding yeast Hop1, we observe no nucleosome binding for ecHOP1 (Fig. 3A). Instead, we find that ecHOP1-wHTH binds to dsDNA (Fig. 3F). While AlphaFold2 modeling suggested ecRED1 might contain six CMs (Fig. 4A), we could only biochemically validate one of these— CM-R-A (Fig. 4C,D). Despite the fact that several of these candidate CMs predictions had very feasible interactions (Fig. 4F). Nonetheless, Red1 in other species (for example, in budding yeast) is highly post-translationally modified (Kar et al, 2022). ecRED1 contains several potential phosphorylatable residues that are in or adjacent to the non-functional CMs (Fig. EV10). These additional putative CMs could be functional under certain circumstances, raising the level of ecHOP1 recruitment at certain loci (Fig. 5).

AlphaFold2 modeling suggested three possible ecHOP1 *cis*-CMs, two in isoform 1 (CM-H1-A and CM-H1-B) and one lower confidence CM in isoform 2 (M-H2-A). We found that both CM-H1-A and CM-H1-B bind to the HORMA domain with equivalent affinity (Fig. 3E), and CM-H2-A does not bind. Bioinformatics

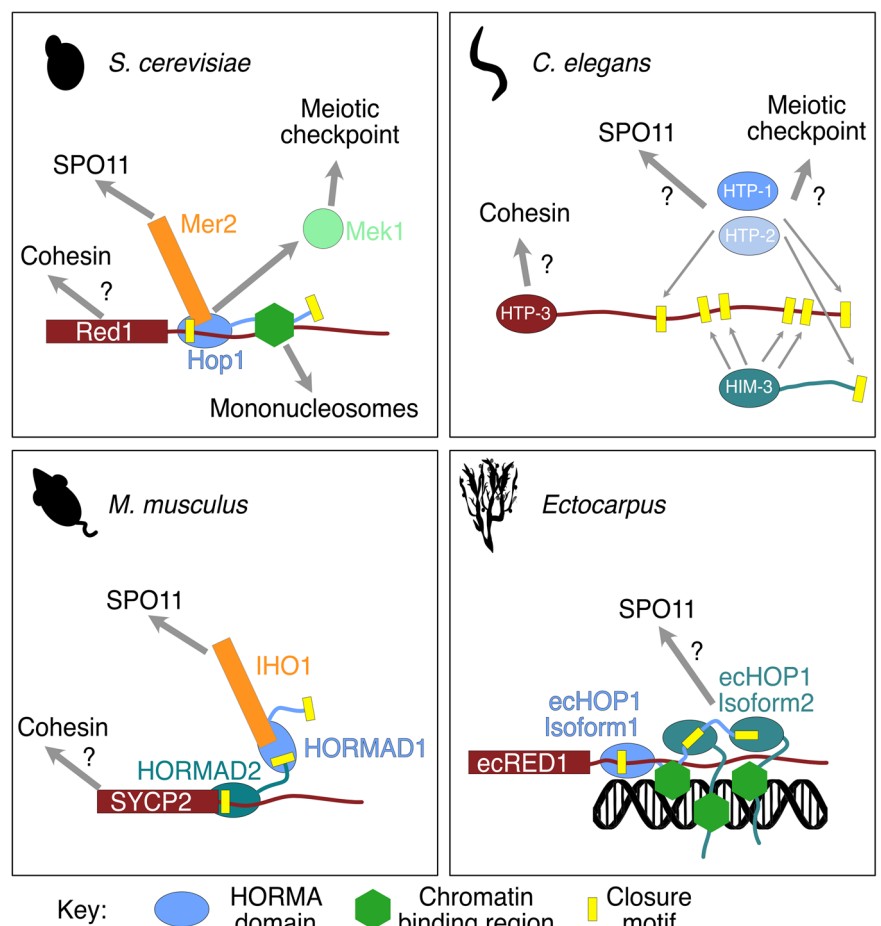

**Figure 5. Comparison of the *Ectocarpus* axial proteins with other model systems.**

In *S. cerevisiae*, Red1 may be recruited via cohesin interactions (Sun et al, 2015). Red1 contains one CM, which can bind to Hop1. Hop1 can be recruited partly independently of Red1 via its CBR binding to nucleosomes (Milano et al, 2024; Heldrich et al, 2022). Hop1 can directly bind to Mer2 when it is bound to the CM of Red1 (Rousova et al, 2021), which in turn regulates Spo11 recruitment. When Hop1 is phosphorylated, it can bind to the checkpoint kinase Mek1 (Carballo et al, 2008; Hollingsworth and Ponte, 1997). In *C. elegans*, there is no Red1 ortholog, instead HTP-3 is recruited to the axis, possibly via cohesin interaction where it can bind to both HIM-3 and HTP-1 and HTP-2 via six CMs (Kim et al, 2014). In mammals, the SYCP2 protein can bind directly to HORMAD2 via a CM (West et al, 2019). SYCP2 might also bind to cohesin, via the CTCF binding site (Li et al, 2020). The CM of HORMAD2 can bind to HORMA1, which in turn can bind to the phosphorylated C-terminus of IHO1 (Stanzione et al, 2016; Dereli et al, 2024). IHO1 is then able to assemble further SPO11-activating factors and recruit/activate SPO11 (Dereli et al, 2024; Laroussi et al, 2023). In *E. siliculosus*, ecRED1 can also bind to ecHOP1-HORMA via CM-R-A. ecRED1 also contains several putative CMs (grayed out boxes), which may be activated via post-translational modifications. Here we show ecHOP1-isoform1 recruiting 2× ecHOP1-isoform2 proteins via CM-H1-A and CM-H1-B. Alternative configurations of ecHOP1 are presumably also possible, which may lead to larger branched assemblies of ecHOP1 on chromatin. In addition to recruitment via ecRED1, ecHOP1 also contains a chromatin-binding region in the form of a dsDNA-binding wHTH domain (green hexagons).

analysis shows that CM-H1-A, CM-H1-B, and CM-R-A in ecRED1 are common to other brown algal lineages (Fig. EV9). ecHOP1 also expresses a second isoform, which contains a HORMA domain, a DNA-binding domain, but no CM. Taken together, this suggests a curious possible arrangement of ecHOP1-isoform1 being recruited to an ecRED1-CM, whereupon it exposes two functional CMs, which could then bind to further HORMA domains (Figs. EV5 and 5). These further HORMA domains could be ecHOP1-isoform1 or isoform2, leading to different possible branched structures. Likewise, isoform2 could be recruited to ecRED1, which would not recruit any further ecHOP1. While the "beads on a string" model for meiotic HORMA domain recruitment has been previously proposed, this type of complex branching assembly of meiotic HORMA domains has previously only been

proposed in nematodes, which contain multiple meiotic HORMA domains (Fig. 5). These potential arrangements, combined with the ecRED1 cryptic CMs, suggest that *Ectocarpus* might utilize varying levels of chromosomal ecHOP1 as a means of regulating DSB formation and the meiotic checkpoint.

This study also highlights the need for careful biochemical validation of computational models. In the case of the HORMA domain to CM interactions, all putative CMs, except CM-H2-A, were predicted to bind to the HORMA with similar confidence by both AlphaFold2 and by AlphaFold3 (although the latter provided consistently lower confidence scores). Close analysis of the side chain interactions for all six of the ecRED1 CMs (CM-R-A; M-R-B to M-R-F) suggested that these interactions are plausible. Despite this, biochemical assays could only validate CM-H1-A, CM-H1-B,

and CM-R-A of the nine predicted CMs. It remains to be seen if these predicted CMs are spurious or if they are activated through post-translational modifications.

Our work raises a number of further questions. Firstly, how is ecRED1 recruited to the axis? Given that ecRED1 appears to be similar to *S. cerevisiae* Red1 and mammalian SYCP2, we might predict that it would bind to cohesin, possibly in a manner similar to binding by CTCF (Li et al, 2020). Secondly, how would ecHOP1-isoform1 be recruited to ecRED1 if it is bound to its own CM? Again, we might speculate that a Pch2/TRIP13 AAA+ ATPase plays a role in opening the HORMA domain and displacing the CM (Raina and Vader, 2020). Thirdly, what are the downstream interactors of ecHOP1? Given the presence of ecSPO11 (Brinkmeier et al, 2022), we speculate that there are additional factors, similar in function to IHO1/Mer2, that help to recruit and activate SPO11. Furthermore, since we note that ecHOP1-HORMA has additional structural elements: the *N*-helix and the $\beta 3''$ hairpin, it might utilize these to bind to a wider range of downstream factors than found in other species. Since the AAA+ ATPase Pch2/TRIP13 interacts with the Hop1 HORMA domain via the *N*-terminus, it is tempting to speculate that the *N*-helix may regulate this process, and further work will be required to test this idea. Finally, what is the identity of the checkpoint kinase in *Ectocarpus*? So far, only Mek1 has been unambiguously assigned as the meiotic checkpoint kinase in budding yeast. Further work remains to determine if *Ectocarpus* also utilizes a meiosis-specific checkpoint kinase, or another cellular kinase fulfills this function.

By tracing the evolutionary lineage of these proteins, our study provides a framework for future research into the molecular evolution of meiosis. It highlights the potential for uncovering regulatory mechanisms in other non-major model organisms that could offer further insights into the origins and diversification of meiosis. In this way, our work not only deepens our understanding of meiotic recombination but also opens new avenues for exploring how evolutionary pressures have shaped the molecular machinery of life over large timescales.

# Methods

### Reagents and tools table

| Reagent/resource | Reference or source | Identifier or catalog number |
|---|---|---|
| **Experimental models** | | |
| BL21(DE3) (*E. coli*) | | |
| BL21(DE3) Artic Express (*E. coli*) | Agilent | #230192 |
| **Recombinant DNA** | | |
| ecHOP1 gBlocks Gene Fragment | This study | |
| ecRED1 gBlocks Gene Fragment | This study | |
| **Antibodies** | | |
| Anti-MBP monoclonal antibody | NEB | E8032L |
| Anti-6xHis mouse monoclonal antibody | Proteintech | 66005-1-IG |

| Reagent/resource | Reference or source | Identifier or catalog number |
|---|---|---|
| Anti-Strep mouse monoclonal antibody | Cube Biotech | 40070 |
| Anti-Strep-tag II rabbit antibody | Abcam | ab76949 |
| Goat anti-mouse HRP | Merck | 401215-2ML |
| Goat anti-rabbit HRP | Merck | 401353-2ML |
| **Oligonucleotides and other sequence-based reagents** | | |
| RT-PCR primers | This study | |
| PCR primers | This study | |
| Fluorescently labeled DNA dsDNA (90 bp) | Altmannova et al, 2021 | https://doi.org/10.1093/nar/gkad175 |
| Fluorescently labeled DNA dsDNA (50 bp) | Altmannova et al, 2021 | https://doi.org/10.1093/nar/gkad175 |
| Fluorescently labeled DNA ssDNA (90 bp) | Altmannova et al, 2021 | https://doi.org/10.1093/nar/gkad175 |
| Fluorescently labeled DNA ssDNA (50 bp) | Altmannova et al, 2021 | https://doi.org/10.1093/nar/gkad175 |
| Fluorescently labeled DNA D-loop | Altmannova et al, 2021 | https://doi.org/10.1093/nar/gkad175 |
| Fluorescently labeled DNA Y-form | Altmannova et al, 2021 | https://doi.org/10.1093/nar/gkad175 |
| Fluorescently labeled DNA HJ | Altmannova et al, 2021 | https://doi.org/10.1093/nar/gkad175 |
| Fluorescently labeled DNA 3' overhang | Altmannova et al, 2021 | https://doi.org/10.1093/nar/gkad175 |
| Fluorescently labeled DNA 5' overhang | Altmannova et al, 2021 | https://doi.org/10.1093/nar/gkad175 |
| **Chemicals, enzymes, and other reagents** | | |
| Thermo Scientific Phusion Flash High Fidelity PCR Master Mix Promotion | ThermoFisher | F548L |
| TURBO DNAse Kit | ThermoFisher | LR-20220214-2 |
| KAPA3G Plant-PCR-Kit | Sigma Aldrich | KK7252 |
| QIAprep Spin Miniprep Kit | QIAgen | 27104 |
| Protease-Inhibitor Mix HP PLUS | SERVA | 39107.02 |
| AEBSF hydrochloride | AppliChem | A1421.0001 |
| DNAseI | BioFroxx | CAS 9003-98-9 |
| Strep-Tactin®XT 4Flow® high-capacity cartridge (5 mL) | IBA Lifesciences | 2-5028-001 |
| MBPTrap HP, 5×5 mL | FisherScientific | 11500664 |
| Cytiva HisTrap™ HP Prepacked Columns | FisherScientific | 11773209 |
| Cytiva HiTrap™ Q Sepharose HP Prepacked Columns | FisherScientific | 11768508 |
| Cytiva HiTrap™ SP Sepharose HP Prepacked Columns | FisherScientific | 11748508 |
| Amicon Ultra 15 mL 30 K MWCO | Sigma Aldrich | UFC903024 |

| Reagent/resource | Reference or source | Identifier or catalog number |
|---|---|---|
| Amicon Ultra 15 mL 10 K MWCO | Sigma Aldrich | UFC901024 |
| MagStrep® Strep-Tactin®XT beads | IBA Lifesciences | 2-5090-010 |
| Pierce™ Streptavidin Magnetic Beads | ThermoFisher | 88816 |
| HiLoad 16/600 Superdex 200 Prep Grade | Cytiva | 28-9893-33 |
| HiLoad 16/600 Superdex 75 Prep Grade | Cytiva | 28-9893-35 |
| Der Blaue Jonas - Single-step Coomassie Blue protein gel dye | Biozol | GRP1 |
| Trans-Blot® Turbo™ RTA Mini Nitrocellulose Transfer Kit, for 40 blots | Bio-Rad | 1704272 |
| Nitrocellulose membrane | GE Healthcare | 10401197 |
| ECL Prime Western Blotting Detection Reagent, for 1000 cm² membrane | GE Healthcare | RPN2232 |
| Slide-A-Lyzer™ Dialysis Devices, 3.5 K MWCO | ThermoFisher | A52967 |
| Superose 6 Increase 5/150 GL | GE Healthcare | 29-0915-97 |
| Albumin, Bovine Serum (BSA), Fraction V, Fatty Acid-Free | Merck | 126575-100GM |
| **Software** | | |
| R version 4.3 | R Core Team (2024) | https://www.r-project.org/ |
| ggplot2 | Springer-Verlag New York | 978-3-319-24277-4 |
| AlphaFold2.3.2 multimer system | | https://doi.org/10.1101/2021.10.04.463034 |
| AlphaFold3 | Google DeepMind | |
| Jalview version 2 | Geoff Barton's group | |
| ChimeraX | https://www.rbvi.ucsf.edu/chimerax/ | https://doi.org/10.1002/pro.4792 |
| PDBePISA | Protein Data Bank in Europe | http://ebi.ac.uk/msd-srv/prot_int/cgi-bin/piserver |
| LigPlot+ | EMBL-EBI | |
| AIUPred3 | | https://aiupred.elte.hu/ |
| IUPred3 | | https://iupred3.elte.hu/ |
| DALI | | http://ekhidna2.biocenter.helsinki.fi/dali/ |
| NetPhos 3.1 | DTU Health Tech | https://services.healthtech.dtu.dk/services/NetPhos-3.1/ |

| Reagent/resource | Reference or source | Identifier or catalog number |
|---|---|---|
| DeepSUMO | | http://deepsumo.renlab.org/server.html |
| IQ-TREE2 | | https://iqtree.github.io/ |
| EMBOSS Needle | EMBL-EBI | https://www.ebi.ac.uk/jdispatcher/psa/emboss_needle |
| **Other** | | |

## Identification of core meiotic genes in *Ectocarpus*

Primary sequences of core meiotic genes from the major model organisms (*Homo sapiens*, *Saccharomyces cerevisiae*, *C. elegans*, and *Arabidopsis thaliana*) were used as query to run BLAST searches on ORCAE (http://bioinformatics.psb.ugent.be/blast/moderated/?project=orcae_EctsiV2) under default settings.

## RNAseq and statistical analysis

RNAseq data representing various life stages (GA, PSP, SP) of the Ec32, Ec25, and Ec17 strains of *Ectocarpus* were obtained from previous studies (Lipinska et al, 2017; Data ref: Lipinska et al, 2017; Lotharukpong et al, 2024; Data ref: Lotharukpong et al, 2024). Reads from the diploid Ec17 strain were used for SP, and Ec32 and Ec25 (haploid strains) were used for GA and PSP stages. Raw counts were transformed into log-transformed expression values as $\log2(\text{TPM}+1)$, where TPM stands for transcripts per million. Boxplots were generated to display the distribution of log-transformed expression levels across three tissue groups: GA (gametophyte), pSP (parthenosporophyte), and SP (sporophyte). To ensure consistency, specific colors were assigned to each group: salmon/magenta for GA, blue/cyan for pSP, and mustard yellow for SP. The boxplots were generated using the ggplot2 package in R. Each gene was visualized individually, with jitter plots added to display individual data points. Custom $y$-axis limits were calculated for each gene, and plots were presented with a gray background and muted axis/grid lines for visual clarity. Boxplots show the median (center line), the interquartile range (IQR) (25th–75th percentiles), and whiskers extending to 1.5× the IQR. Outliers beyond the whiskers are shown as individual points. This approach was intentionally chosen to standardize the visual structure across genes with differing expression magnitudes. Pairwise comparisons between the three tissue groups (GA vs. pSP, pSP vs. SP, and GA vs. SP) were performed using two-sided $t$ tests. An overall analysis of variance test was conducted to determine if there were significant differences among the three groups for each gene. $p$ Values were annotated on the boxplots, and statistical significance was assessed between groups. Brackets were manually added to indicate significant differences between tissue groups, with the corresponding $p$ values displayed adjacent to the brackets. Each gene was plotted using 9–12 replicates for GA and PSP samples (from haploid Ec32 and Ec25) and 3 replicates for SP (from diploid Ec17). All statistical analyses were conducted using R version 4.3 with a significance threshold set at $p < 0.05$.

## RNA extraction from brown algal tissue

Freshly cultivated *Ectocarpus* strains were prepared as previously described (Coelho et al, 2012). Algae were flash frozen in liquid nitrogen placed into individual 1.5 mL Eppendorf tubes, and homogenized into a fine powder with tissue grinding pestles (BioEcho Life Sciences). Throughout the lysis, the tubes were routinely placed back into LN. In all, 50 μL of preheated CTAB3 Extraction Buffer (100 mM Tris-HCl, pH 8.0, 1.4 M NaCl, 20 mM EDTA, 2% plant RNA isolation aid (PVP), 2% CTAB, 1% β-mercaptoethanol) was added to each sample and vortexed for 1 min, and quickly centrifuged for a 1 s pulse. The lysed solution was then transferred to a RNAse-free 2 mL tube (Eppendorf). In all, 50 μL of achloroform/isoamylalcohol (24:1) solution was added, vortexed vigorously, and then subjected to centrifugation at 10,000 rpm for 15 min at 4 °C, following the segregation of the upper aqueous layer into a new RNAse-free 2 mL tube. This process was repeated twice. On the third cycle, the upper aqueous phase was transferred into a new 1.5 mL RNAse-free tube (Eppendorf) with 100 μL of 7.5 M LiCl and a 1% (*v/v*) β-mercaptoethanol. The samples were agitated vigorously and RNA precipitated O/N at −20 °C. The samples were centrifuged at 15 K rpm for 1 min at 4 °C, and the supernatant was discarded. The pellet was washed with 1 mL of ice-cold 70% ethanol, then centrifuged again at 5 K rpm for 10 min at 4 °C. The residual ethanol was removed from the pellet, and the samples were left to dry for no more than 5 min. The pellets were dissolved in RNAse-free $H_2O$, and RNA concentration was determined with a Nanodrop spectrophotometer. DNAse treatment was completed with the TURBO DNAse Kit (Thermo-Fisher) following the manufacturer's protocol. RNA aliquots at 500 ng were prepared and stored in −80 °C for future use.

## RT-PCR to validate ecHOP1 transcriptional isoforms

Tissue samples (in triplicate) of Ec17 (SP life stage) unilocular sporangia (uniloc sets 1–3), SP vegetative tissue (without unilocular sporangia), and Ec32 (pSP) vegetative tissue (without unilocular sporangia) were utilized for RT-PCR to determine RNA expression of both ecHOP1 isoforms. RNA extraction followed a protocol previously optimized for *Ectocarpus* tissue (Cossard et al, 2022). Complementary primers were designed with consideration for the difference in exons between each isoform, as well as control ecHOP1 primers that are specific to regions that encapsulate 100% conservation between both isoforms (Table EV1). Dynein was utilized as a control since it is a housekeeping gene. cDNA lacking reverse transcriptase (-RT) and Kapa polymerase (-Kapa) of pSP vegetative tissue were also used for PCR negative controls. Reactions were run under the "Touchdown" protocol to increase primer specificity. Following amplification, 60 DNA Loading Dye (ThermoFisher) was added into each sample at a 1:5 ratio and loaded onto a 2% agarose gel. The gel ran at 80 V for approximately 1 h.

## Phylogenetic tree construction

Full-length protein sequences of ecHOP1 and meiotic HORMA homologs were retrieved from *Ectocarpus siliculosus* and a selection of representative species across major eukaryotic supergroups from Uniprot, including model organisms (*Homo sapiens* (HORMAD1/HORMAD2); *Arabidopsis thaliana* (ASY1); *Oryza sativa* (PAIR2); *Saccharomyces cerevisiae* (HOP1), *Schizosaccharomyces pombe* (HOP1), *Caenorhabditis elegans* (HTP-1/HTP-2/HTP-3/HIM-3)), one haptophyte (*Emiliania huxleyi* (putative HOP1 ortholog)), one cryptophyte (*Salpingoeca rosetta* (putative HOP1 ortholog)), and one chlorophyte (*Chlamydomonas reinhardtii* (HOP1 ortholog). Sequences were aligned using MAFFT (v7.505) with the L-INS-i algorithm for high accuracy. The alignment was manually inspected and trimmed using trimAl (v1.4.rev22) with the -automated1 setting to remove poorly aligned regions. Phylogenetic inference was conducted using IQ-TREE2 (v2.4.0). The best-fitting substitution model was selected using ModelFinder, which identified the LG + G4 model as the optimal choice based on the Bayesian Information Criterion (BIC). The maximum-likelihood tree was constructed with ultrafast bootstrap approximation using 1000 replicates (-bb 1000) to assess node support. The final tree was visualized using iTOL v6 (Interactive Tree Of Life), where nodes were annotated based on species taxonomy and known eukaryotic supergroups. Bootstrap values were mapped onto the internal nodes, and color-coded clade highlights were used to distinguish major evolutionary lineages.

## Cloning of the StrepII-ecHOP1-HORMA, His$_6$SUMO-ecHOP1-HORMA, His$_6$SUMO-ecHOP1-wHTH, and His$_6$MBP-ecRED1-CMs

Sequences of full-length *Ectocarpus* ecHOP1 and ecRED1 were codon optimized for *E. coli* recombinant protein expression and ordered as gBlocks Gene Fragments (GenScript). All desired full-length and truncations were amplified by Phusion Flash High-Fidelity PCR using gene-specific primers (Table EV2; IDT DNA), and subsequent assembling via Gibson Assembly Cloning within the desired pCOLI- expression vectors (Altmannova et al, 2021). Expression constructs are listed in Table EV3.

## Overexpression and purification of ecHOP1-HORMA domain

ecHOP1-*HORMA* was expressed as a 3C HRV cleavable *N*-terminal StrepII-tag fusion in BL21(DE3) competent cells. The cultures were grown at 37 °C until an OD$_{600}$ of 0.6 was achieved. Protein expression was induced with 500 μM IPTG at 16 °C for approximately 16 h. The cells were centrifuged at 15,000 rpm for 10 min, and the pellets were resuspended in cell lysis buffer (100 mM KH$_2$PO$_4$ pH 7.3, 250 mM NaCl, 1 mM MgCl$_2$, 10 mM arginine, 10 mM glutamate, 10 mM imidazole, 0.1% Triton X-100). The cells were then lysed using an EmulsiFlex C3 (Avestin) in the presence of SERVA protease (25 μg/mL), AEBSF protease (25 μg/mL), and DNAseI (10 μg/mL) (ThermoFisher) and cleared through ultra-centrifugation at 40,000 rpm for 1 h at 4 °C (Beckman Coulter). The supernatant was then passed through a 5 mL StrepTactinXT 4 flow column (IBA Lifesciences), previously equilibrated in wash buffer (100 mM KH$_2$PO$_4$ pH 7.3, 250 mM NaCl, 1 mM EDTA, 1 mM MgCl$_2$, 1 mM TCEP, 10 mM arginine, 10 mM glutamate). The column was washed with an ATP wash buffer (100 mM KH$_2$PO$_4$ pH 7.3, 250 mM NaCl, 1 mM EDTA, 1 mM MgCl$_2$, 1 mM TCEP, 10 mM arginine, 10 mM glutamate, 1 μM ATP), followed by a high salt wash buffer (100 mM KH$_2$PO$_4$ pH 7.3, 800 mM NaCl, 1 mM EDTA, 1 mM MgCl$_2$, 1 mM TCEP, 10 mM arginine, 10 mM glutamate) to eliminate non-specific contaminants such as

chaperones and DNA. The protein was then eluted with elution buffer (100 mM $KH_2PO_4$ pH 7.3, 800 mM NaCl, 1 mM EDTA, 1 mM $MgCl_2$, 1 mM TCEP, 10 mM arginine, 10 mM glutamate, 50 mM biotin), concentrated on an Amicon concentrator (30 MWCO; ThermoFisher), and loaded onto a Superdex 200 16/600 (Cytiva) pre-equilibrated in wash buffer. Biochemically pure fractions were pooled, flash frozen in $N_{2(l)}$, and stored at −80 °C for future use.

ecHOP1-HORMA was also expressed as a 3C HRV cleavable *N*-terminal His$_6$SUMO-tag fusion in BL21(DE3) competent cells. The cultures were grown at 37 °C until an $OD_{600}$ of 0.6 was achieved. Protein expression was induced with 500 μM IPTG at 16 °C for approximately 16 h. The cells were centrifuged at 15,000 rpm for 10 min, and the pellets were resuspended in cell lysis buffer (100 mM $KH_2PO_4$ pH 7.3, 250 mM NaCl, 1 mM $MgCl_2$, 10 mM arginine, 10 mM glutamate, 10 mM imidazole, 0.1% Triton X-100). The cells were then lysed using an EmulsiFlex C3 (Avestin) in the presence of SERVA protease (25 μg/mL), AEBSF protease (25 μg/mL), and DNAseI (10 μg/mL) (ThermoFisher) and cleared through ultracentrifugation at 40,000 rpm for 1 h at 4 °C (Beckman Coulter). The supernatant was then passed through a 5 mL HisTrap column (Cytiva), previously equilibrated in wash buffer (100 mM $KH_2PO_4$ pH 7.3, 250 mM NaCl, 1 mM EDTA, 1 mM $MgCl_2$, 10 mM arginine, 10 mM glutamate, 10 mM imidazole). The column was extensively washed in wash buffer, then the protein was then eluted with elution buffer (100 mM $KH_2PO_4$ pH 7.3, 800 mM NaCl, 1 mM EDTA, 1 mM $MgCl_2$, 10 mM arginine, 10 mM glutamate, 250 mM imidazole), and buffer exchanged via dialysis (3.5 MWCO; ThermoFisher) in low salt buffer (100 mM $KH_2PO_4$ pH 7.3, 50 mM NaCl, 1 mM EDTA, 1 mM $MgCl_2$, 1 mM TCEP, 10 mM arginine, 10 mM glutamate). The protein was then loaded onto a HiTrap S column (Cytiva) pre-equilibrated in low salt buffer. A stepwise gradient with high salt buffer (100 mM $KH_2PO_4$ pH 7.3, 1 M NaCl, 1 mM EDTA, 1 mM $MgCl_2$, 1 mM TCEP, 10 mM arginine, 10 mM glutamate) was used to elute the protein and non-specific contaminants from the column. The flow through, wash, and elution gradients up to 25% were pooled, concentrated on an Amicon concentrator (10 MWCO; ThermoFisher), flash frozen in $N_{2(l)}$, and stored at −80 °C for future use.

## Overexpression and purification of ecHOP1-wHTH domain (WT and R432A)

ecHOP1-wHTH was expressed as a 3C HRV cleavable *N*-terminal His$_6$SUMO-tag fusion in BL21(DE3) competent cells. The cultures were grown at 37 °C until an $OD_{600}$ of 0.6 was achieved. Protein expression was induced with 500 μM IPTG at 16 °C for approximately 16 h. The cells were centrifuged at 15,000 rpm for 10 min, and the pellets were resuspended in cell lysis buffer (100 mM $KH_2PO_4$ pH 7.3, 250 mM NaCl, 1 mM $MgCl_2$, 10 mM arginine, 10 mM glutamate, 10 mM imidazole, 0.1% Triton X-100). The cells were then lysed using an EmulsiFlex C3 (Avestin) in the presence of SERVA protease (25 μg/mL), AEBSF protease (25 μg/mL), and DNAseI (10 μg/mL) (ThermoFisher) and cleared through ultra-centrifugation at 40,000 rpm for 1 h at 4 °C (Beckman Coulter). The supernatant was then passed through a 5 mL HisTrap column (Cytiva), previously equilibrated in wash buffer (100 mM $KH_2PO_4$ pH 7.3, 250 mM NaCl, 1 mM EDTA, 1 mM $MgCl_2$, 10 mM arginine, 10 mM glutamate, 10 mM imidazole). The column was

extensively washed in wash buffer, then the protein was then eluted with elution buffer (100 mM $KH_2PO_4$ pH 7.3, 800 mM NaCl, 1 mM EDTA, 1 mM $MgCl_2$, 10 mM arginine, 10 mM glutamate, 250 mM imidazole), and buffer exchanged via dialysis (3.5 MWCO; ThermoFisher) in low salt buffer (100 mM $KH_2PO_4$ pH 7.3, 50 mM NaCl, 1 mM EDTA, 1 mM $MgCl_2$, 1 mM TCEP, 10 mM arginine, 10 mM glutamate). The protein was then loaded onto a HiTrap S column (Cytiva) pre-equilibrated in low salt buffer. A stepwise gradient with high salt buffer (100 mM $KH_2PO_4$ pH 7.3, 1 M NaCl, 1 mM EDTA, 1 mM $MgCl_2$, 1 mM TCEP, 10 mM arginine, 10 mM glutamate) was used to elute the protein and non-specific contaminants from the column. The flow through, wash, and elution gradients up to 25% were pooled, concentrated on an Amicon concentrator (10 K MWCO; ThermoFisher), flash frozen in $N2_{(l)}$, and stored at −80 °C for future use.

## Overexpression and purification of ecRED1 CMs

All ecRED1-CMs were expressed as a 3C HRV cleavable *N*-terminal His$_6$MBP-tag fusion in BL21(DE3) Arctic Express competent cells. The cultures were grown at 30 °C until an $OD_{600}$ of 0.6 was achieved. Protein expression was induced with 500 μM IPTG at 12 °C for approximately 16 h. The cells were centrifuged at 15,000 rpm for 10 min, and the pellets were resuspended in cell lysis buffer (50 mM HEPES pH 7.5, 300 mM NaCl, 1 mM TCEP, 0.1% Triton X-100). The cells were then lysed using an EmulsiFlex C3 (Avestin) in the presence of SERVA protease (25 μg/mL), AEBSF protease (25 μg/mL), and DNAseI (10 μg/mL) (Thermo-Fisher) and cleared through ultracentrifugation at 40,000 rpm for one hour at 4 °C (Beckman Coulter). The supernatant was then passed through a 5 mL HisTrap column (Cytiva), previously equilibrated in wash buffer (50 mM HEPES pH 7.5, 300 mM NaCl, 1 mM EDTA, 1 mM TCEP). The column was extensively washed in wash buffer, then the protein was then eluted with elution buffer (100 mM $KH_2PO_4$ pH 7.3, 800 mM NaCl, 1 mM EDTA, 1 mM $MgCl_2$, 10 mM arginine, 10 mM glutamate, 250 mM imidazole), and concentrated on an Amicon concentrator (10 MWCO; ThermoFisher) and loaded onto a Superdex 200 16/600 (Cytiva) pre-equilibrated in wash buffer. Biochemically pure fractions were pooled, flash frozen in $N_{2(l)}$, and stored at −80 °C for future use.

## Isothermal titration calorimetry

ITC experiments were completed with a MicroCal PEAQ-ITC (Malvern Panalytical). All protein samples were extensively dialyzed in ITC buffer (50 mM HEPES pH 7.5, 150 mM NaCl, 1 mM EDTA) in Slide-A-Lyzer™ Dialysis Devices, 3.5 K MWCO (ThermoFisher) at 4 °C O/N. Final concentrations were validated for both ecHOP1-HORMA and all CMs, at 25 and 350 μM, respectively. A ratio of 1:40 was used due to the predicted low binding affinity between the HORMA domain and CMs, as also recommended by the equipment manufacturer. The titrants (CMs, individually) were injected into the titrand (ecHOP1-HORMA) at a starting injection of 0.4 μL, followed by 12 × 3 μL injection volumes, with 180 s spacing, and spinning at 500 rpm, while maintaining a temperature of 25 °C. Control runs were completed with the same parameters, whether the titrant or titrand was measured during the addition of the ITC buffer. Buffer runs were also extensively evaluated to ensure minimal non-specific heat signals would not be

acquired. All experiments were completed in triplicate to ensure reproducibility.

## Pull-down assays

Strep-TactinXT pull-down assays were performed with purified ecHOP1-HORMA domain with a Twin-Strep-tag and His$_6$-MBP-tagged ecRED1 putative CM peptides. In all, 250 µL reactions with 20 µg of each bait and prey protein in wash buffer (100 mM Tris pH 8.0, 300 mM NaCl, 1 mM EDTA, 1 mM TCEP) were incubated with 5 µL of magnetic Strep-TactinXT beads (IBA Lifesciences) for 20 min at 4 °C with periodic gentle mixing. Beads were then washed two times with 500 µL of wash buffer. Bound proteins were eluted by resuspending the beads in 125 µL of elution buffer (100 mM Tris pH 8.0, 300 mM NaCl, 1 mM EDTA, 1 mM TCEP, 5% glycerol, 50 mM biotin) and subsequent incubation for 15 min at 4 °C with gentle mixing throughout, repeating this process three times. To analyze the remaining proteins bound to the beads, a denaturing elution was performed by resuspending the beads in 30 µL of SDS loading dye and boiling for 5 min at 95 °C. Samples for SDS-PAGE were prepared at a 2:1 ratio of sample and SDS loading dye, then boiled for 5 min at 95 °C. The gels were visualized with Coomassie staining (Der Blaue Jonas, Biozol), and western blot, using a 1:10,000 dilution of anti-MBP-tag antibody in 5% milk in 1× TBST, pH 7.6 (New England BioLabs), washed in 1× TBST, pH 7.6, followed by secondary anti-mouse antibody at 1:20,000 for 20 min RT in 3% BSA in 1x TBST, pH 7.6 (Merck).

## Electrophoretic mobility shift assays

Triplicate EMSAs were completed at a constant nucleic acid (167 bp Widom sample, dsDNA, ssDNA) concentration of 50 nM and seven samples of each protein ranging from 25 nM to 2 µM. The samples were incubated for approximately 30 min at 4 °C, then loaded onto a 0.8% agarose gel in 0.2% TBE Buffer. The gel was run for 2 h at 4 °C, not exceeding a power of 60 V, then post-stained with SYBRGold (Invitrogen). Gels were imaged using a ChemiDocMP (Bio-Rad Inc.). Nucleic acid depletion in each lane was quantitated with the imager software, using measurements of triplicate of nucleic acid alone for each individual gel as a baseline. Binding curves were fitted using Prism software and the following algorithm ($Y = \mathrm{Bmax} * Xh/(K_\mathrm{D}h + Xh)$).

## AlphaFold modeling

Modeling of protein structures was performed with the AlphaFold multimer system (version 2.3.2) (Jumper et al, 2021; Evans et al, 2021) with two multimer predictions per model and the Amber relaxation procedure applied to the best model (Case et al, 2023). While modeling the ecHOP1-HORMA together with ecpreRED1 to discover the ecRED1 CM, several putative ecRED1 CMs appeared in the predicted models. First, the HORMA domain of ecHOP1 was modeled together with full-length ecRED1, followed by the model generations of the ecHOP1-HORMA with ecRED, however, with a truncation of the *C*-terminal CC region to mitigate low confidence model predictions due to limitations in CC domain predictions. After evaluating all putative CMs consistently appeared within the same boundaries in these models as well, further predictions were generated solely with the truncations of each putative CM in conjunction with the HORMA domain.

## CM motif identification and analysis

The phylogenetic tree used to compare the evolution of HOP1 and RED1 CMs in brown algae was modeled after a reference previously reported.

Protein sequences of different brown algae species were retrieved through the BLAST interface of the Phaeoexplorer brown algal genome database (https://phaeoexplorer.sb-roscoff.fr/home/), using the brown algal protein sequences of ecHOP1 and ecRED1 as query. Results of the BLAST search were aligned using MAFFT MSA tools (Katoh et al, 2002). Manual curation of the MSA was performed using JalView (Waterhouse et al, 2009), and alignments for each putative CM were determined by *N*- and *C*-terminal lysine and/or arginine residues deemed essential for CM boundaries based on interactions observed between the HORMA domain and CMs in structural models generated with Alpha Fold 2.3.

Protein models predicted with Alpha Fold 2.3 were analyzed with PyMOL and ChimeraX (Meng et al, 2023) in regards of conserved structures and residues in the interaction site between ecHOP1 and ecRED1 and their truncations. Eighteen models were chosen for further evaluation of the interaction between ecHOP1-HORMA and the putative ecRED1 CMs. Models were selected according to their rank and confidence of the model, which is indicated by the PAE scores. The best ranking models generally have higher confidence and less clashes between residues, hence these were selected preferentially. Due to the varying numbers of models that show interactions between ecHOP1-HORMA and the putative ecRED1 CMs, the number of evaluated models per CM differs. Interactions between ecHOP1 and ecRED1 were determined by visualization of hydrogen bonds and polar contacts in a radius of 4 Å around the CM residues to ecHOP1-HORMA. Furthermore, the presence of hydrophobic interactions, π–stacking, or other interactions like π–Met, π–Thiol, and π–cation/anion were investigated. Predicted H$^-$ bonds and salt bridges were further confirmed by PDBePISA (Krissinel and Henrick, 2007), which predicts interactions between two chains of a model computationally, and through LigPlot+ to both interpret and visualize interactions (Laskowski and Swindells, 2011).

## Protein disorder prediction

AIUPred3 web server (https://iupred3.elte.hu/) was utilized to predict disorder scores along the full-length ecHOP1 and ecRED1 protein sequences (Erdős and Dosztányi, 2024). IUPred was run in long disorder prediction mode; for ecRED1, the CM regions were compared to disorder profiles to confirm their placement within predicted unstructured regions.

## In silico PTM predictions of ecHOP1 and ecRED1

Phosphorylation sites were predicted using both full-length sequences of ecHOP1 and ecRED1 with NetPhos 3.1 (https://services.healthtech.dtu.dk/services/NetPhos-3.1/), identifying serine, threonine, and tyrosine residues with high modification probability based on default thresholds. SUMOylation predictions, including candidate SUMO interaction motifs (SIMs) and lysine residues for SUMO conjugation, were obtained using DeepSUMO (http://deepsumo.renlab.org/server.html). Predicted sites were mapped onto AlphaFold3 structural models of ecHOP1 and ecRED1 to assess spatial accessibility. While multiple SUMOylation sites were identified, attempts to model SUMO binding at these

locations did not yield stable complexes, suggesting possible context-dependent interactions or low structural compatibility in isolation. Phosphorylation predictions indicated numerous potential sites on both proteins, supporting a role for extensive PTM regulation. Only high-confidence predictions, as defined by the respective software thresholds, were retained for further analysis and cross-referenced for consistency.

## Circular dichroism

CD experiments were collected on a JASCO J-815 Circular Dichroism Spectropolarimeter (JASCO Inc., Japan). In all, 20 μM ecHOP1-wHTH WT and R432A samples were measured in CD and Tm Buffer (10 mM $KH_2PO_4$, pH 7.4, 30 mM NaCl) in a 1 mm quartz cuvette. CD spectra were collected from 190 to 260 nm, with 6 accumulation trials.

## SEC-MALS

Triplicates at 1 mg/mL (approximately 50 μM) were loaded onto a Superose 6 5/150 analytical size exclusion column (Cytiva) equilibrated in SEC-MALS buffer (50 mM HEPES, pH 7.5, 150 mM NaCl, 1 mM TCEP) attached to a 1260 Infinity II LC System (Agilent). MALS was completed with a Wyatt DAWN detector attached in line with the size exclusion column. Calibration and normalization were completed with monomeric BSA at 1 mg/mL in the SEC-MALS buffer.

## Data availability

This study includes no data deposited in external repositories.

The source data of this paper are collected in the following database record: biostudies:S-SCDT-10_1038-S44319-025-00605-3.

## Peer review information

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

## Acknowledgements

We would like to thank Remy Luthringer for supplying the various *Ectocarpus* tissue for these studies and for providing microscopic images of the various life stages of *Ectocarpus*. We also would like to thank Fabian Haas for their support with RNAseq analysis and assistance in generating boxplots, and Josué Barrera-Redondo for their contributions to the initial identification of ecHOP1 and ecRED1 in the *Ectocarpus* genome. We thank Gerben Vader (Princess Maxima Cancer Centre, Utrecht) for comments on the manuscript. JRW is supported by the Max Planck Society and the German Research Foundation (Grant Number WE 6513/2-1). SMC is supported by the Max Planck Society, The European Research Council (Grant Number 864038), The Moore Foundation (Grant 11489), and the Bettencourt Foundation.

## Author contributions

**Emma I Kane**: Formal analysis; Supervision; Validation; Investigation; Visualization; Methodology; Writing—original draft; Project administration; Writing—review and editing. **Lioba S Trefs**: Data curation; Formal analysis; Validation; Investigation; Methodology. **Lena Eckert**: Investigation. **Susana M Coelho**: Resources; Formal analysis; Supervision; Funding acquisition; Project administration; Writing—review and editing. **John R Weir**: Conceptualization; Formal analysis; Supervision; Funding acquisition; Investigation; Visualization; Writing—original draft; Project administration; Writing—review and editing.

Source data underlying figure panels in this paper may have individual authorship assigned. Where available, figure panel/source data authorship is listed in the following database record: biostudies:S-SCDT-10_1038-S44319-025-00605-3.

## Funding

## Disclosure and competing interests statement

The authors declare no competing interests.

# Expanded View Figures

---

**Figure EV1.   MSA ecHOP1 transcriptional isoforms.**

(A) (top) Genome annotation figure of the two ecHOP1 isoforms from ORCAE. (bottom) Alignment between the two transcriptional isoforms of ecHOP1 (Ec-07_005800.1 for isoform #1 and Ec-07_005800.2 for isoform #2) show high conservation between the two. Interestingly, there is an insertion between a.a. 495-514 in isoform #2 (blue), as well as a *C*-terminal extension following a.a. 574 in isoform #1 (orange). Although the insertion in isoform #2 does not computationally demonstrate to influence the protein structure, the extension in isoform #1 contains the two putative CMs. The DNA sequences for each isoform were acquired through ORCAE, and the translated sequences were utilized for MSA. The MSA was generated with JalView and aligned with MAFFT using default settings. (B) Topology map of ecHOP1 HORMA domain, colored as in Fig. 2. (C) AlphaFold2 model of ecHOP1-HORMA alongside experimentally determined HORMA domain structures from nematodes (*C. elegans*; HTP-3 with HIM-2 CM (PDB: 4TZM)) and mammalians (*H. sapiens* HORMAD1 bound in *cis* to HORMAD1 CM (PDB: 8J69)). (D) MSA of the brown algal HOP1 orthologs, highlighting the novel β3″ region (residues 94-107 in ecHOP1). A color gradient from light to dark was implemented to indicate the percentage of conservation. The alignment was generated using MAFFT and curated in Jalview.

---

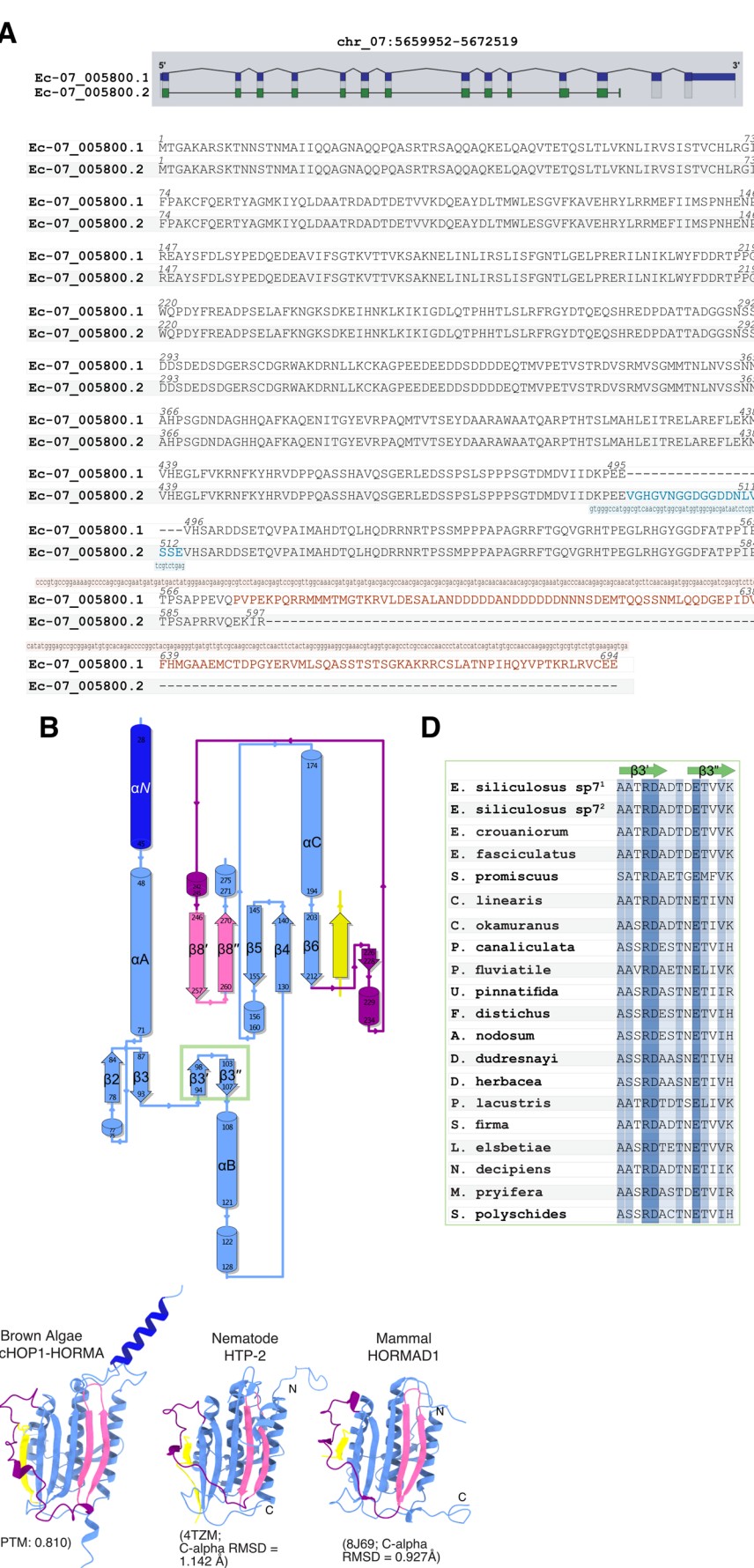

A

chr_07:5659952-5672519

Ec-07_005800.1
Ec-07_005800.2

Ec-07_005800.1    MTGAKARSKTNNSTNMAIIQQAGNAQQPQASRTRSAQQAQKELQAQVTETQSLTLVKNLIRVSISTVCHLRGI 73
Ec-07_005800.2    MTGAKARSKTNNSTNMAIIQQAGNAQQPQASRTRSAQQAQKELQAQVTETQSLTLVKNLIRVSISTVCHLRGI 73

Ec-07_005800.1    FPAKCFQERTYAGMKIYQLDAATRDADTDETVVKDQEAYDLTMWLESGVFKAVEHRYLRRMEFIIMSPNHENP 146
Ec-07_005800.2    FPAKCFQERTYAGMKIYQLDAATRDADTDETVVKDQEAYDLTMWLESGVFKAVEHRYLRRMEFIIMSPNHENP 146

Ec-07_005800.1    REAYSFDLSYPEDQEDEAVIFSGTKVTTVKSAKNELINLIRSLISFGNTLGELPRERILNIKLWYFDDRTPPG 219
Ec-07_005800.2    REAYSFDLSYPEDQEDEAVIFSGTKVTTVKSAKNELINLIRSLISFGNTLGELPRERILNIKLWYFDDRTPPG 219

Ec-07_005800.1    WQPDYFREADPSELAFKNGKSDKEIHNKLKIKIGDLQTPHHTLSLRFRGYDTQEQSHREDPDATTADGGSNSS 292
Ec-07_005800.2    WQPDYFREADPSELAFKNGKSDKEIHNKLKIKIGDLQTPHHTLSLRFRGYDTQEQSHREDPDATTADGGSNSS 292

Ec-07_005800.1    DDSDEDSDGERSCDGRWAKDRNLLKCKAGPEEDEEDDSDDDDEQTMVPETVSTRDVSRMVSGMMTNLNVSSNN 365
Ec-07_005800.2    DDSDEDSDGERSCDGRWAKDRNLLKCKAGPEEDEEDDSDDDDEQTMVPETVSTRDVSRMVSGMMTNLNVSSNN 365

Ec-07_005800.1    AHPSGDNDAGHHQAFKAQENITGYEVRPAQMTVTSEYDAARAWAATQARPTHTSLMAHLEITRELAREFLEKM 438
Ec-07_005800.2    AHPSGDNDAGHHQAFKAQENITGYEVRPAQMTVTSEYDAARAWAATQARPTHTSLMAHLEITRELAREFLEKM 438

Ec-07_005800.1    VHEGLFVKRNFKYHRVDPPQASSHAVQSGERLEDSSPSLSPPPSGTDMDVIIDKPEE--------------- 495
Ec-07_005800.2    VHEGLFVKRNFKYHRVDPPQASSHAVQSGERLEDSSPSLSPPPSGTDMDVIIDKPEEVGHGVNGGDGGDDNLV 511
                                                                            gtgggccatggcgtcaacggtggcgatggtggcgacgataatctcgtg

Ec-07_005800.1    ---VHSARDDSETQVPAIMAHDTQLHQDRRNRTPSSMPPPAPAGRRFTGQVGRHTPEGLRHGYGGDFATPPIP 565
Ec-07_005800.2    SSEVHSARDDSETQVPAIMAHDTQLHQDRRNRTPSSMPPPAPAGRRFTGQVGRHTPEGLRHGYGGDFATPPIP 584
                  tcgtctgag
                  cccgtgccggaaaagcccagcgacgaatgatgactatgggaacgaagccgtcctagacgagtccgcgttggcaaacgatgatgatgacgccaacgacgacgacgacgatgacaacaacagcgacgaaatgaaccaacagagcagcaacatgcttcaacaagatggcgaaccgatcgacgtcttc

Ec-07_005800.1    TPSAPPEVQPVPEKPQRRMMMTMGTKRVLDESALANDDDDDANDDDDDDNNNSDEMTQQSSNMLQQDGEPIDV 638
Ec-07_005800.2    TPSAPRRVQEKIR--------------------------------------------------------- 597
                  catatgggagccgcgagatgtgcacagaccccggctacgagaggtgatgttgtcgcaagccagctcaactctcaactagcgggaagtggaaacgtaggtgcagcctcgccaccaaccctatccatcagtatgtgccaaccaagagtgtgcgtgtctgtgaagagtgaa

Ec-07_005800.1    FHMGAAEMCTDPGYERVMLSQASSTSTSGKAKRRCSLATNPIHQYVPTKRLRVCEE 694
Ec-07_005800.2    -------------------------------------------------------

B

C

Brown Algae
ecHOP1-HORMA

(AF2; PTM: 0.810)

Nematode
HTP-2

(4TZM;
C-alpha RMSD =
1.142 Å)

Mammal
HORMAD1

(8J69; C-alpha
RMSD = 0.927Å)

D

| | β3' | β3'' |
|---|---|---|
| E. siliculosus sp7[1] | AATRDADT | DETVVK |
| E. siliculosus sp7[2] | AATRDADT | DETVVK |
| E. crouaniorum | AATRDADT | DETVVK |
| E. fasciculatus | AATRDADT | DETVVK |
| S. promiscuus | SATRDAET | GEMFVK |
| C. linearis | AASRDADT | NETIVN |
| C. okamuranus | AASRDADT | NETIVK |
| P. canaliculata | ASSRDEST | NETVIH |
| P. fluviatile | AAVRDAET | NELIVK |
| U. pinnatifida | AASRDAST | NETIIR |
| F. distichus | ASSRDEST | NETVIH |
| A. nodosum | ASSRDEST | NETVIH |
| D. dudresnayi | ASSRDAAS | NETIVH |
| D. herbacea | ASSRDAAS | NETIVH |
| P. lacustris | AATRDTDT | SELIVK |
| S. firma | AATRDADT | NETVVK |
| L. elsbetiae | AASRDTET | NETVVR |
| N. decipiens | AATRDADT | NETIIK |
| M. pryifera | AASRDAST | DETVIR |
| S. polyschides | ASSRDACT | NETVIH |

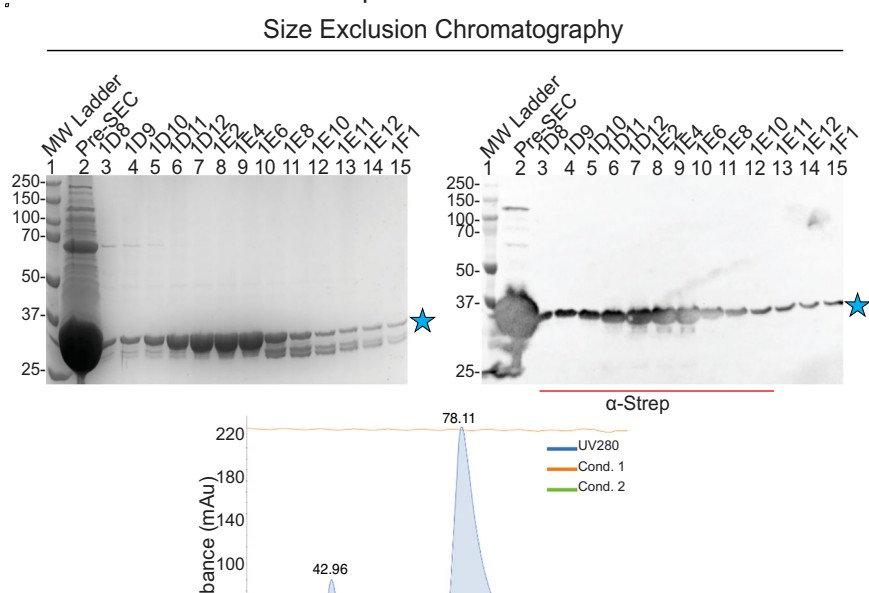

**Figure EV2. Purification of ecHOP1-HORMA.**

Purification of 2xStrepII-ecHOP1-HORMA. The Coomassie-stained gel image and western blot shown here also appear in Fig. 2A.

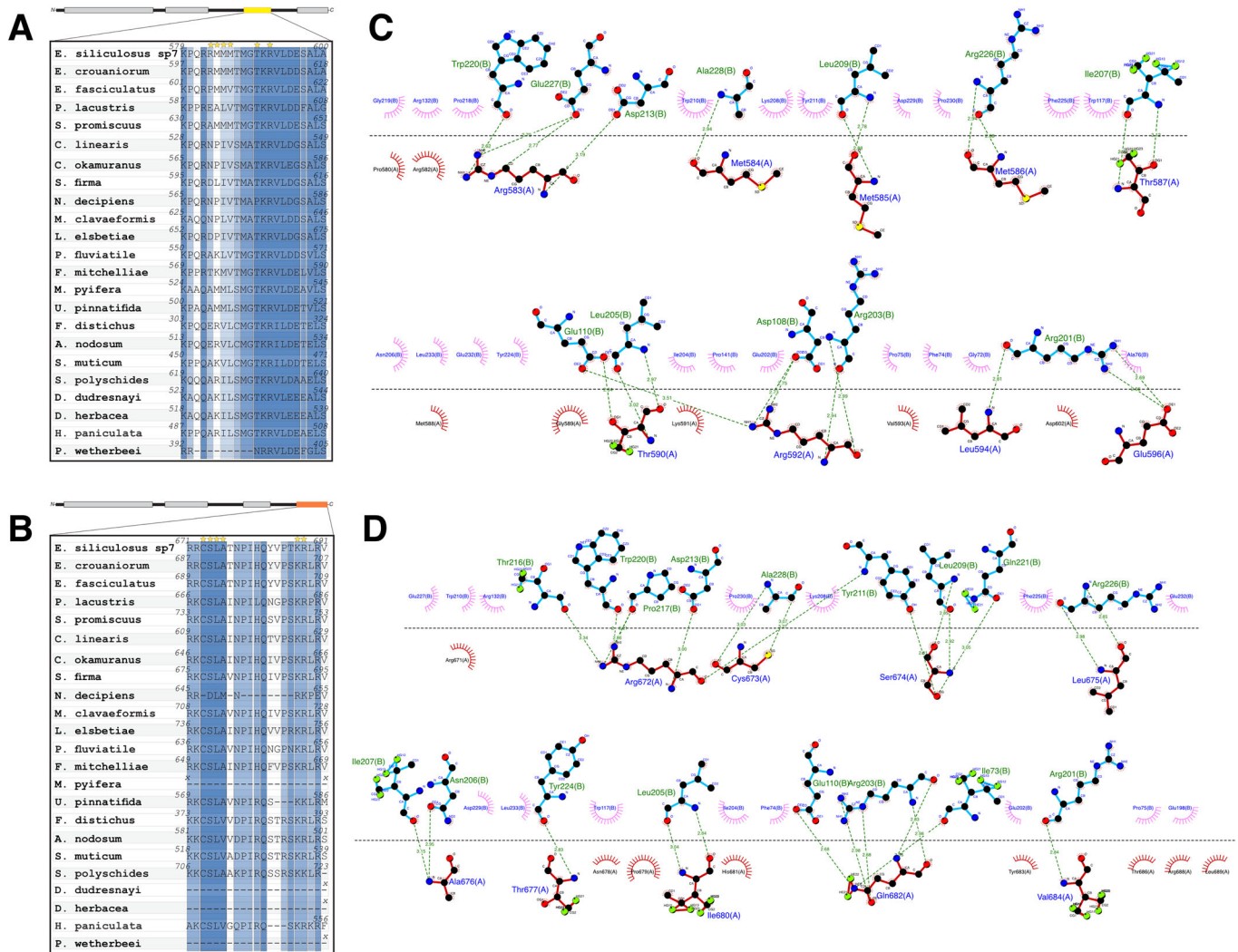

**Figure EV3. Details of CM-H1-A and CM-H1-B.**

(A) MSA of CM-H1-A with other sequences from within Stramenopiles. Residues predicted to be involved in the interaction with the HORMA domain are indicated with yellow stars. (B) MSA of CM-H1-B with other sequences from within Stramenopiles. Residues predicted to be involved in the interaction with the HORMA domain are indicated with yellow stars. (C) LigPlot analysis of the top-rated (based on iPTM) prediction of CM-H1-A with ecHOP1-HORMA. (D) LigPlot analysis of the top-rated (based on iPTM) prediction of CM-H1-B with ecHOP1-HORMA.

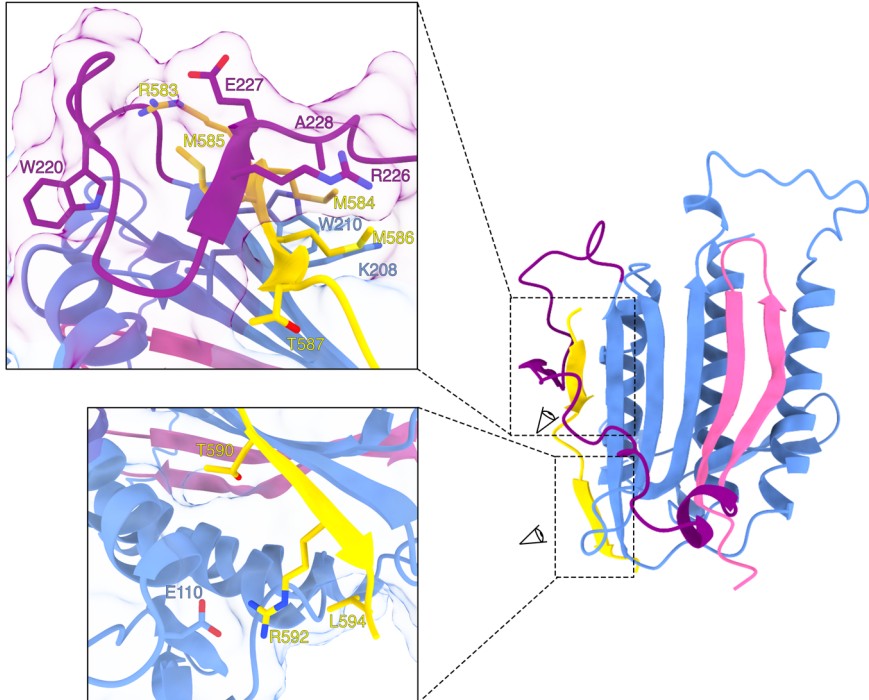

**Figure EV4. Structural analysis of the predicted interaction of CM-H1-A with ecHOP1-HORMA.**

AlphaFold2 model of ecHOP1-HORMA in complex with CM-H1-A (yellow), highlighting key intermolecular interactions. (top) Close-up of the CM-H1-A residues R583, M584, M585, and M588 engaging ecHOP1-HORMA core residues K208 and W210 (blue), as well as the conserved ecHOP1-HORMA residues within the safety belt region R226, W220, E227, and A228 (purple). R583 is predicted to form side-chain H-bonds with E227 and the main chain of W220, while the methionine cluster forms hydrophobic contacts with surrounding residues. (bottom) Additional contacts between CM-H1-A residues T590, R592, and L594 (yellow) and ecHOP1-HORMA residue E110 (blue), which is conserved and situated outside the safety belt.

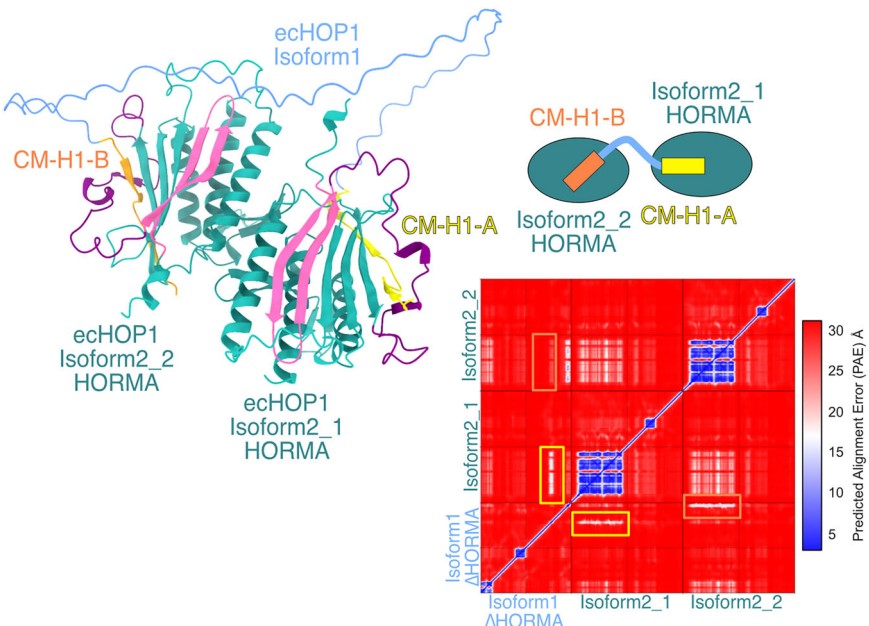

**Figure EV5. ecHOP1-isoform 1 could bind two ecHOP1-isoform 2.**

AlphaFold2 model of a complex of ecHOP1-isoform1 (with the *N*-terminal HORMA domain removed; residues 562–694) (blue; CM-H1-A, yellow; CM-H1-B, orange) and two copies of full-length ecHOP1-isoform2. The model is colored as elsewhere, but the ecHOP1-HORMA-isoform2 domains are colored in teal. In the PAE plot, the CMs are highlighted with the colors as in the cartoon.

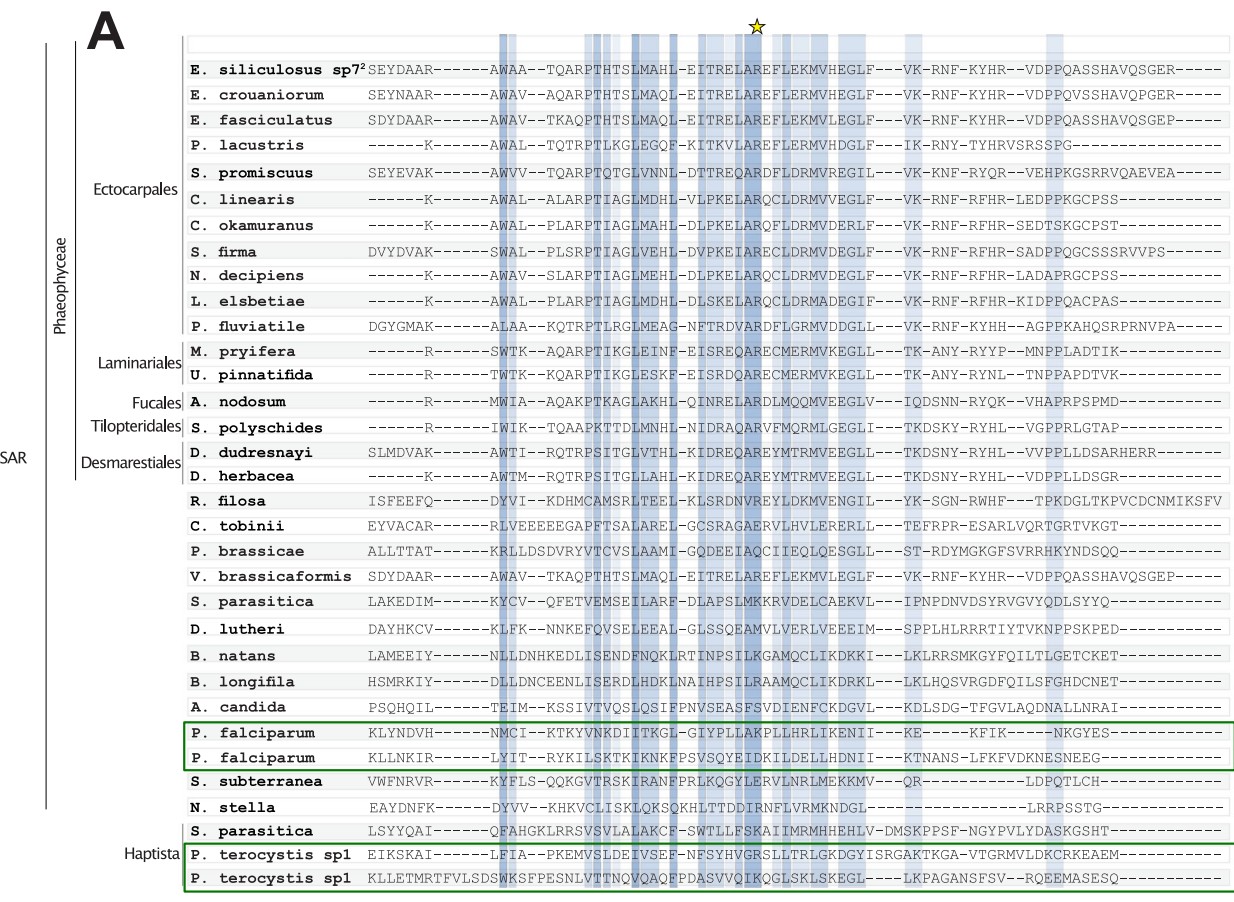

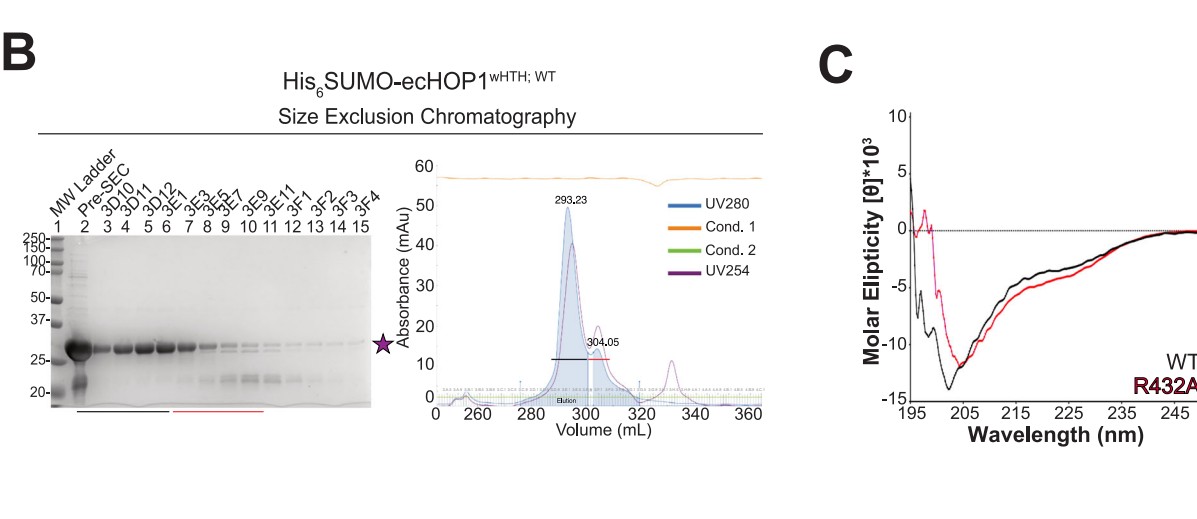

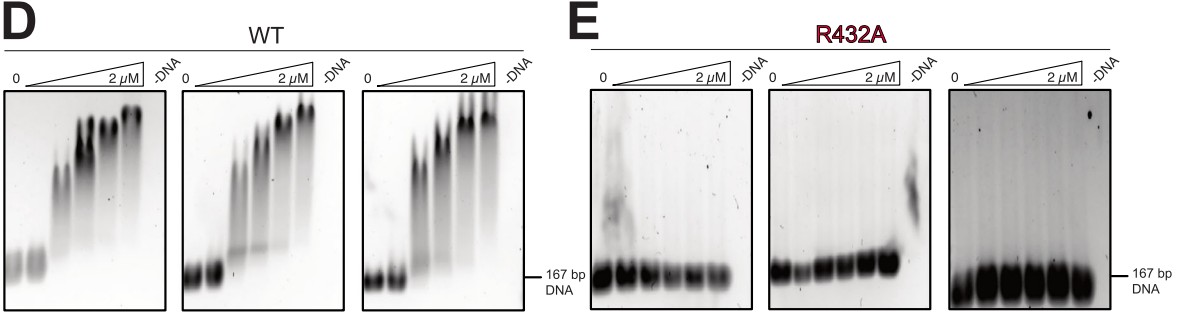

◄ **Figure EV6. ecHOP1-wHTH and DNA binding properties.**

(A) MSA of HOP1 wHTH within SAR and Haptista clades. The two species with twin wHTH domains are highlighted (green boxes). (B) Purification of the His$_6$SUMO-ecHOP1-wHTH. (C) Circular dichroism spectra of ecHOP1-wHTH WT (black) and R432A mutant (red). The wHTH domain is composed of three α-helices, a β-loop, and a flexible wing, typically producing minima near 208, 217, and 222 nm. Similar spectra for WT and R432A indicate that the mutation does not significantly alter the secondary structure or overall fold of the domain. (D) Triplicate EMSAs of WT ecHOP1-wHTH on 167 bp dsDNA. The first of these EMSAs is shown in Fig. 3I. (E) Triplicate EMSAs of ecHOP1-wHTH R432A mutant on 167 bp dsDNA. The first of these EMSAs is shown in Fig. 3I.

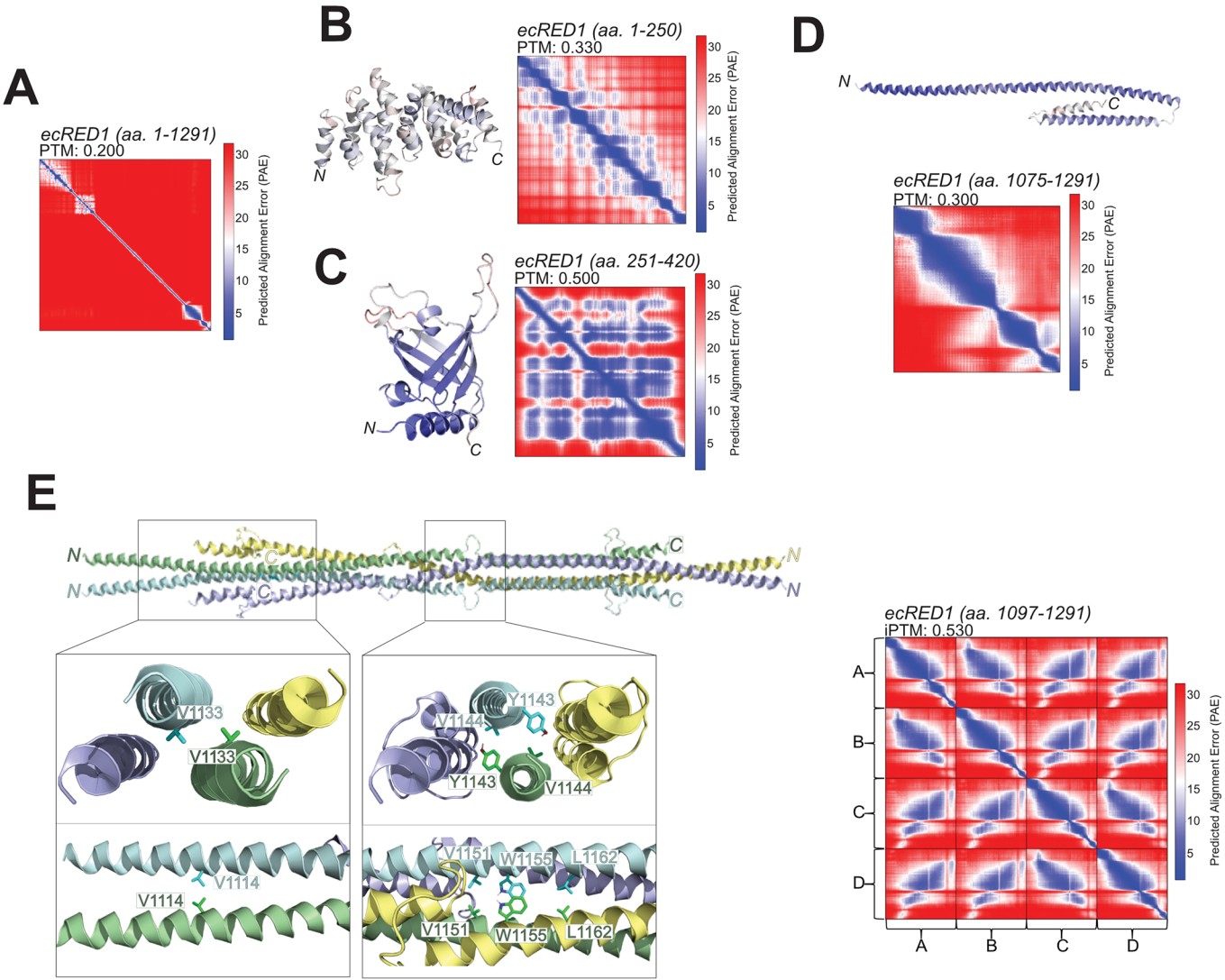

**Figure EV7. ecRED1 structure modeling and validation.**

(A–D) AlphaFold2 model of ecRED1 domains with PAE plots. (E) AlphaFold2 model of ecRED1 CC domain as a tetramer with PAE plot.

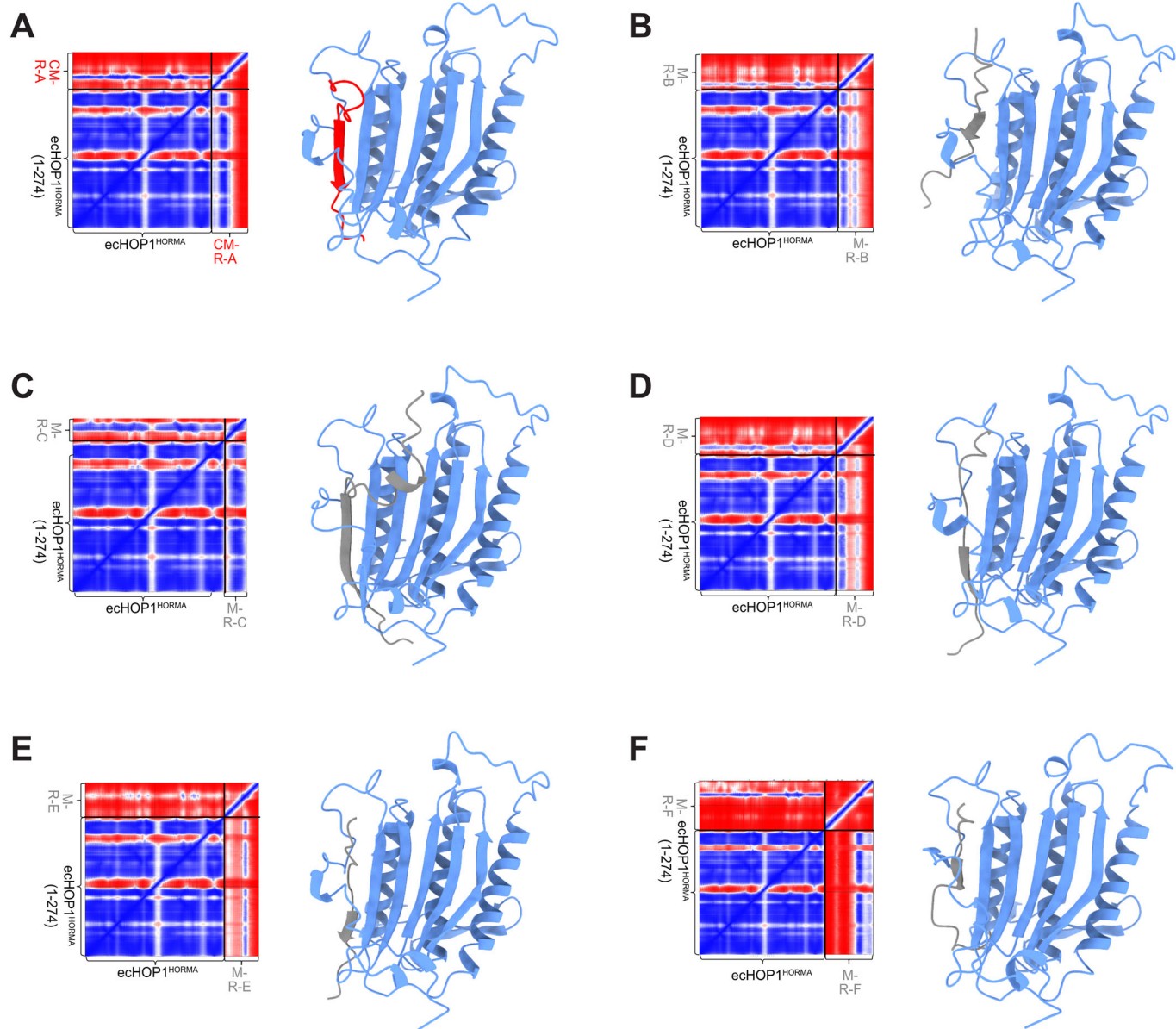

**Figure EV8. PAE plots and models of ecHOP1-HORMA with ecRED1 candidate closure motifs.**

(**A**) PAE plot, colored as in Fig. 4E. Cartoon representation of ecHOP1-HORMA in blue, CM-R-A in red. (**B–F**) PAE plot, colored as in Fig. 4E. Cartoon representation of ecHOP1-HORMA in blue, M-R-B to M-R-F in gray.

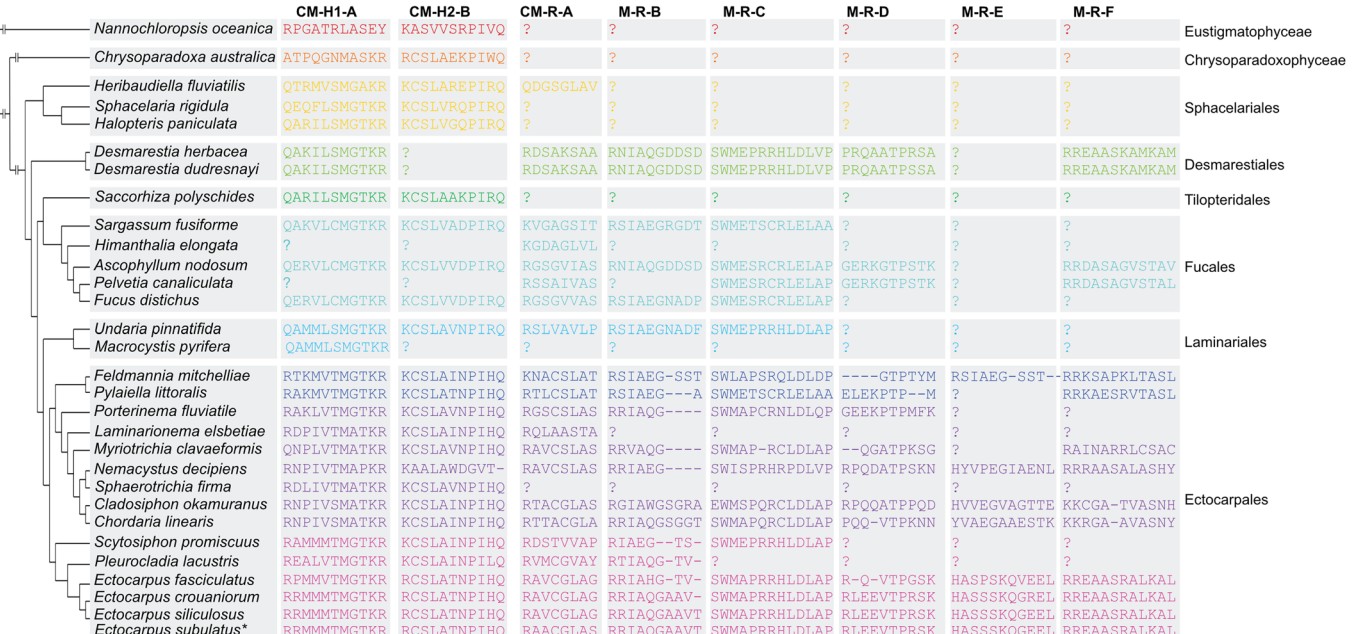

**Figure EV9. Closure motif phylogeny.**

MSA and phylogenetic analysis of brown algal HOP1 and RED1 orthologs reveal a lineage-specific rise of CMs. Absence of specific CMs in certain organisms is due to poor genome annotations or lack of conservation in relevant regions. In Ectocarpales, ecHOP1 and ecRED1 CMs are highly conserved, with additional ecRED1 CMs emerging in Desmarestiales. HOP1 orthologs in Eustigmatophyceae and Chrysoparadoxophyceae show two well-conserved CMs, suggesting an early addition of CMs for regulatory functions. The alignment was generated using MAFFT and curated in Jalview to identify conserved lysine and arginine residues defining CMs.

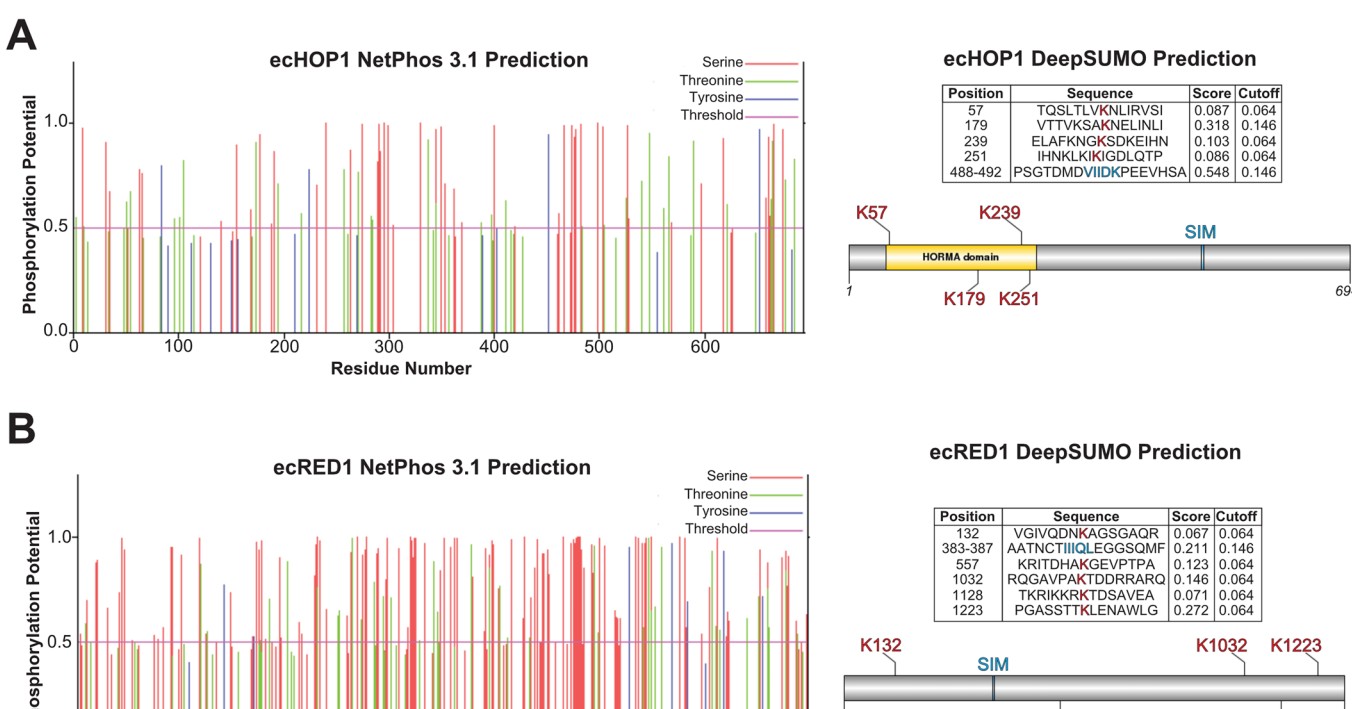

**Figure EV10. Potential post-translational modifications on ecHOP1 and ecRED1.**

(A) Predicted phosphorylation and SUMOylation sites mapped onto the AlphaFold3 model of ecHOP1. (B) Predicted phosphorylation and SUMOylation sites mapped onto the AlphaFold3 model of ecRED1. Phosphorylation sites (serine, threonine, tyrosine) and candidate SUMOylation motifs, including SUMO interaction motifs (SIMs), were predicted in silico using NetPhos 3.1 and DeepSUMO, respectively. Several predicted phosphorylation sites cluster near putative or non-functional CMs in ecRED1. Although multiple SUMOylation sites were identified, structural modeling did not support stable SUMO binding, suggesting possible regulation dependent on additional in vivo factors.

