## [Peer Review File · EMBO Reports]

Characterization of meiotic axis proteins in the model brown alga *Ectocarpus*

Emma Kane, Lioba Trefs, Lena Eckert, Susana Coelho, and John Weir

Corresponding author(s): John Weir (john.weir@tuebingen.mpg.de), Susana Coelho (susana.coelho@tuebingen.mpg.de)

Review Timeline:

Transfer Date:	6th Jun 25
Editorial Decision:	29th Jul 25
Revision Received:	7th Aug 25
Accepted:	26th Sep 25

Transaction Report: This manuscript was transferred to EMBO reports following peer review at The EMBO Journal.

Referee #2:

In this manuscript, Kane et al have identified and characterised axial element components (ecHOP1 and ecRED1) from the meiotic chromosome axes of brown algae. Importantly, they find that these brown algae proteins using the same principles as in other model organisms (e.g. mice and yeast), indicating that these underlying mechanisms have been conserved throughout evolution. Overall, this study has been performed carefully, the data are of high quality, and it provides important evolutionary insights into meiosis. It is certainly of interest to the broad readership of EMBO Journal, so I am happy to recommend that it is accepted for publication subject to addressing the following minor comments and corrections.

1. In the introduction, general statements are made regarding axial proteins recruiting HORMA proteins through closure motifs in yeast, mammals and plants. I don't think that this has been shown conclusively in all cases. For example, as far as I'm aware, the only evidence regarding SYCP2 is a Y2H interaction between a long fragment and HORMAD2. I think some caveats should be added, and ideally specific references of what has been shown, to ensure that readers are aware of the uncertainties.

2. Figure 2C - The overall AF2 model is likely to be correct given the similarity to other HORMA proteins. Why did the authors choose CM-1 rather than CM-2 as the bound closure motif? It would be helpful to see details of the side-chain interactions at the closure motif interface (in the model and in the experimental structure), along with an alignment, to see whether they appear plausible. Also, what do the authors make of the N-helices in the AF2 models? Do the sequences look as though they should be helical, are they largely hydrophilic or do they have a potential heptad repeat that could imply involvement in a coiled-coil structure?

3. Figure 3B - Whilst I am confident with the interpretation that the protein is a monomer, the MW is stated with false precision at 34.09 kDa in the figure and 34.01 kDa in the main text. The MW fit across the curve shows deviate that far exceeds the stated precision, and other limitations mean that MALS data are typically accurate to between 5-10 kDa (i.e. more than sufficient to tell whether it is a monomer or dimer etc). I recognise that an error range is shown, but such false precision may mislead readers who are unfamiliar with the technique. I would suggest that the MW should be stated as approximately 34 kDa. Also, it would be helpful if it were plotted on a linear scale so the range of MWs across the peak could be interpreted. Further, the theoretical MW of a monomer should be stated in the figure legend.

4. Figure 3B and S3 - I am not convinced that the 57 kDa species is a real second species as the absorbance barely shows a peak above the baseline, so may simply represent noise. Did the authors try analysing material at different concentrations to see whether there is concentration-dependent assembly into a dimer? If they want to include this, they should integrate the two peaks separately and state the percentage of material by mass in each peak so the relative amount is clear to readers (I imagine the second peak will be <1%). I think the AF2 modelling of dimers is very speculative as AF2 comes up with the best model assuming that the given oligomer is correct, and I have seen similar and far higher levels of confidence for completely spurious associations. It is good that they performed the pull-downs upon deleting the N-helices. On this basis of the data shown, I would say after investigating whether it could form a dimer, they have determined that it seemingly remains monomeric. I suggest editing the text to prevent readers from misinterpreting these points (e.g. I would remove the Af2 dimer modelling and simply state that they tested whether it could form a dimer with/without the N-helix and found no evidence for self-association).

5. Figure 3C-E,K - I would suggest some caution in interpreting the AF2 models as it tends to hallucinate interfaces for well-known interacting proteins, meaning that most conserved regions of unstructured sequence are predicted to bind as closure motifs when modelled together. The key discriminator is whether the side-chain interactions are favourable. Hence, I suggest showing panels of the predicted side-chain interfaces of the putative CMs in their complexes in comparison with the experimentally determined interfaces (that could be shown in figure 2). The authors should discuss in the text whether the predicted interfaces appear plausible. In relation to this, it is good that the authors tested these interactions in pull-downs (with targeted mutations) and by ITC. It is notable that CM1-2 had a confident AF2 complex but failed to show binding by ITC - might this be an example of such an hallucination?

6. Figure 3K - where is the evidence for this trans-interaction given that CM1-2 failed to bind to the HORMA domain? As above, I would only present AF2 models if they are supported by experimental evidence.

7. Figure 3J - This is a pedantic point, but it is incorrect to say that $K_d=0$, as it is not determined to be zero by the assay. Instead, ITC performed using these protein concentrations with associated limitations etc failed to detect an interaction, and the experimental setup would not detect a low affinity interaction (e.g. > 1mM). Instead, I would state $K_d=n.d.$ (not determined).

8. Line number 291 - SYCP2 has not been found to form homotetramers. SYCP3 forms mostly homotetramers and may also form 2:2 complexes with SYCP2.

9. Figure S5 - Has ecRED1 been shown to form a tetramer? Has the AF2 model been validated experimentally, and do the side-chain interactions appear plausible? Models of coiled-coils are very often incorrect even when there is high confidence, so I would only show such models if there is appropriate validation.

10. Figure 5E - As previously, side-chain interactions of the putative closure motif should be shown to determine whether they appear realistic.

Referee #3:

General Summary

This manuscript explores how meiotic axis formation is regulated in the model brown alga *Ectocarpus*, focusing on two proteins predicted to function analogously to the well-characterized Hop1 and Red1 in other eukaryotes:

1. ecHOP1: A HORMA-domain protein with two isoforms. Structural and biochemical data reveal that ecHOP1 harbors a winged helix-turn-helix (wHTH) domain that binds dsDNA, and up to three predicted closure motifs (CMs), two of which (CM1-1 and CM2-1) are shown to bind in cis to its own HORMA domain.

2. ecRED1: Contains several predicted CMs (the best validated is CM1) capable of binding the ecHOP1 HORMA domain. This suggests ecRED1 can recruit ecHOP1, likely assembling the chromosome axis during meiosis.

In addition to biochemical pull-downs (including mutated motifs), ITC, and SEC-MALS, the authors present transcriptomic data revealing that these genes are selectively expressed in sporophyte meiotic tissues but are absent in parthenosporophytes undergoing a form of non-reductive (apo)meiosis. These findings position *Ectocarpus* as a strategically important model for studying meiotic diversity, expanding our evolutionary understanding beyond classical plant, animal, and fungal systems.

Opinion & Principal Significance

A. Mechanistic Insight into Meiotic Axis Assembly: The discovery that ecHOP1 has multiple closure motifs and a robust dsDNA-binding domain highlights a flexible mechanism for chromosome binding. This flexibility could regulate break formation or checkpoint signaling in ways distinct from yeast or vertebrates.

B. Evolutionary Perspective: *Ectocarpus* diverged from canonical models; nonetheless,

ecHOP1/ecRED1 share fundamental HORMA and Red1-like features. This supports the idea that meiosis evolved only once (in the LECA) but diversified via lineage-specific rearrangements (e.g., additional/alternative motifs).

C. Comparative Genomics: By examining ecHOP1 and ecRED1 alongside known homologs (Hop1, Red1, SYCP2, HORMAD1/2), the manuscript broadens our phylogenetic understanding of meiotic proteins. It paves the way for future comparative studies on additional Stramenopiles.

D. Study Limitations: while the biochemical data and AlphaFold predictions are compelling, *in vivo* localization or functional assays (e.g., immunofluorescence, gene tagging) are still lacking. Developing such tools in *Ectocarpus* would further strengthen the conclusions on ecHOP1's recruitment, post-translational modifications, and overall axis architecture.

Points of Attention

A. Functional Validation of Predicted Closure Motifs - The authors have tested all predicted CMs in ecHOP1. They show that CM1-1 and CM2-1 indeed bind the HORMA domain, whereas CM1-2 does not. Thus, for ecHOP1, these motifs appear fully explored. By contrast, ecRED1 has multiple CMs predicted by AlphaFold, but only CM1 (and perhaps weak binding by CM4) is clearly demonstrated to interact with ecHOP1 *in vitro*. The other predicted sites remain unverified. It would strengthen the paper to clarify why these putative CMs did not show binding (e.g., tested but negative? conditionally active?). Also could the authors show the AF2/AF3 data for these CMs and summarise their iPTM values and any other confidence values to give the reader an indication of how relevant each of the CMs is? In general it is not entirely clear to me what the exact reasoning is to classify each CM beyond maybe a conserved arginine/lysine. Could the authors expand on this and make it more complete? Suggestion is to expand the discussion to explain whether the unvalidated ecRED1 motifs are false positives, require post-translational modifications, or might be engaged under specific biological conditions. In case of PTMs these might be tested using AF3?

B. Temporal Regulation and Post-Translational Modifications - Yeast Hop1 and mammalian HORMADs are regulated by phosphorylation or SUMOylation to tune break formation and checkpoint responses. The data here indicate that ecRED1 and ecHOP1 might also be modified, yet no direct evidence (phospho-mutants, MS data) is provided. Even a short pilot experiment or extended discussion of potential kinase/ATPase homologs (Mek1, Pch2) would help the reader appreciate the possible regulatory circuitry.

C. Alternative (Apo)Meiosis in *Ectocarpus* - The authors note that ecHOP1/ecRED1 expression is absent in parthenosporophytes, which undergo a non-reductive (apo)meiosis. This difference is intriguing for understanding how the chromosome axis

might diverge in non-canonical meiosis. The manuscript would benefit from acknowledging whether other axis-like factors could be present in pSP stages, or if those algae bypass the need for a classical axis. Even speculation would clarify broader evolutionary ramifications.

Minor Concerns & Editorial Points

A. Protein Nomenclature & Consistency

- a. Standardize ecHOP1 vs. EcHop1, etc.
- b. Clarify your notation for isoforms (e.g., "ecHOP1^{Iso1}," "ecHOP1^{Iso2}," or "ecHOP1^{HORMA¹⁻²⁷⁴}") to ensure clarity when discussing distinct constructs.

B. RNA Expression Analysis - Briefly mention whether you used housekeeping genes or how you handled normalization. Also clarify the statistical tests (e.g., T-tests) for differential expression.

C. Contextual/Line-Specific Items

- Line 46-49: Strengthen the justification for using a stramenopile model and how that addresses "large evolutionary context."
- Line 128-136: Reads more like an introduction than results. Consider relocating.
- Minor Typos:
 - o Line 55: remove "the" before "gametophyte."
 - o Line 195-200: check for doubled sentences.
 - o Line 344: missing comma after "motif."
 - o Line 620: "lised" → "lysed."
 - o Line 971: "SCYP2" → "SYCP2."
- ORCAE: Define the acronym or website context upon first mention.

D. Hop1-Related Points

- Figure 2E (percent identity/similarity) may be less informative than a proper phylogeny showing ecHOP1 clustering with known Hop1 orthologs (and an outgroup of another HORMA protein).
- If two isoforms exist in Ectocarpus, clarify whether other brown algae also share these predicted isoforms, suggesting deeper evolutionary conservation.
- In Supplementary Figure 1B, highlight which species have the extra β -sheet region ($\beta 3'/\beta 3''$) - include representatives of other eukaryotic clades. If it's truly unique to brown algae, a brief speculation on function beyond other interactors would be valuable (perhaps an AF-based interactor screen would be an option?).

E. wHTH Domain

- The multi-sequence alignment (Figure 4G) might be more approachable if limited to Stramenopiles (alignment seems not very strong with only three very distant organisms in it). If referencing widely divergent taxa, clarify their potential structural differences (single vs. double wHTH domains - which of the two domains is used?).
- Consider performing an AF3 run with ecHOP1 bound to dsDNA to refine the interaction model (any clashes or additional insights?).

F. Alignments

Throughout the manuscript various multiple alignments are shown. It would be good to use a consistent coloring scheme and perhaps also a consistent tool to generate the alignments. My advice is to the common jalview scheme that many are used to - but this is up to the authors of course. I am mentioning jalview as the use of such tools make sure all residues that are supposed to be in the same column are properly aligned - now in suppl figure on the CMs in ecRed1 this is not the case. Also the coloring scheme here doesn't help to assess how conserved positions really are. This would help this type of analysis considerably as I think this is an excellent figure in general. Sidenote - did the authors consider adding in disorder prediction to assess whether these CMs are in fact available to bind to ecHop1 and are not in a globular region?

Additional Suggestions for Improving the Study

- A. In Vivo Localization Studies - while challenging, immunofluorescence or gene tagging would directly confirm how and where ecHOP1 and ecRED1 localize in meiotic cells.
- B. Expanded DNA-Binding Specificity - assessing whether ecHOP1 prefers certain DNA structures (e.g., supercoiled, specific sequences, nucleosomal) could illuminate how break hotspots might be designated in brown algae.
- C. Post-Translational Regulation - since PTMs are pivotal in many meiotic systems, systematically probing for phosphorylated or SUMOylated residues (and their functional consequences) would be a logical next step (some of it might be show already with AF3).

Overall Recommendation

This manuscript provides novel insights into how a meiotic axis may be assembled in a non-traditional eukaryotic model and lays the groundwork for future comparative approaches. The authors have tested all predicted CMs in ecHOP1 (clarifying which ones do or do not bind), but several ecRED1-predicted motifs remain unconfirmed. With some additional detail on unvalidated CMs, a deeper discussion of possible post-translational modifications, and some future follow-up in vivo confirmation, this study will represent a valuable contribution to our understanding of meiosis across the eukaryotic tree.

Referee #4:

The manuscript "Characterization of meiotic axis proteins in the model brown alga *Ectocarpus*" by Kane et al. describes the first characterization of the meiotic axis proteins HOP1 and RED1 in the brown algae *Ectocarpus*, an emerging model system for the study of meiosis. HOP1 is a member of the broad HORMA domain protein family of meiotic recombination regulators, and RED1 is a so-called "axis core" protein putatively involved in organizing meiotic chromosomes along with meiosis-specific cohesin complexes. Briefly, the authors demonstrate that HOP1 and RED1 are expressed specifically in the meiotic life cycle stage; identify two isoforms of HOP1 and dissect their domain architectures; dissect the "closure motifs" in both HOP1 and RED1 that bind the HOP1 HORMA domain, and show that the HOP1 winged helix-turn-helix domain binds DNA.

Overall, there is interesting data here, but I don't feel that there is sufficient novelty to warrant publication in *The EMBO Journal*. The novel features of these proteins, including their putative possession of multiple closure motifs, mainly rest on inconclusive AlphaFold predictions, and are not sufficiently justified or supported by their experimental data.

Specific points:

The authors should make some attempt to discern which isoform of ecHOP1 is more prevalent. Does their PCR indicate that isoform 1 is more prevalent? Is there evidence of which ecHOP1 isoform predominates from the RNA-Seq dataset the authors draw from in Figure 1?

Related to the above, in Figure 2B (Gel 3), the 276 bp band is difficult to visualize. A darker exposure or adjusted contrast may help clearly show amplification from both ecHOP1 isoforms.

I don't see the gray arrow in figure 2C as stated in the text.

Lines 157-183: The authors seem to have deleted any description of how the different putative closure motifs in the two ecHOP1 isoforms were identified from this section. Did they perform AlphaFold with both isoforms? Did the closure motifs show confident interactions with the HORMA domain? I know some of these questions are addressed later,

but it's jarring that they are not addressed in this section, especially given the phrasing of the last sentence in this section.

Supplementary figure 3: please report AlphaFold ipTM (interface pTM) values, which report on the confidence of complex formation. (Currently pTM scores are noted, which is not the same).

Also related to Supplementary Figure 3: The HOP1 dimer models shown in panels B and D are not consistent with one another. Given this inconsistency, and the authors' failure to confirm HOP1 dimerization experimentally, I suggest leaving out this inconclusive data.

For the three ecHOP1 closure motifs identified by the authors, does the sequence alignment shown in Figure 3F match up with the register predicted by AlphaFold for these motifs' binding to the ecHOP1 HORMA domain?

Line 301: reference to figure 5D should be Figure 5B.

Where is the evidence that AlphaFold 2 identified six putative closure motifs in ecRED1? There should be supplemental figure panels showing more detailed AlphaFold results supporting Figure 5B.

More broadly, if the authors' data indicate that there is only one functional closure motif in ecRED1 (which it does), why even mention the other five? This gets to a larger issue running throughout the paper: describing inconclusive/low confidence AlphaFold predictions, then showing that experiments cannot confirm these predictions. Given the possibility that AlphaFold is simply wrong, why are these predictions reported if they can't be supported with experimental evidence?

Kane *et al.* "Characterization of meiotic axis proteins in the model brown alga *Ectocarpus*" Rebuttal

Referee #2:

In this manuscript, Kane *et al.* have identified and characterised axial element components (ecHOP1 and ecRED1) from the meiotic chromosome axes of brown algae. Importantly, they find that these brown algae proteins using the same principles as in other model organisms (e.g. mice and yeast), indicating that these underlying mechanisms have been conserved throughout evolution. Overall, this study has been performed carefully, the data are of high quality, and it provides important evolutionary insights into meiosis. It is certainly of interest to the broad readership of EMBO Journal, so I am happy to recommend that it is accepted for publication subject to addressing the following minor comments and corrections.

We appreciate the reviewer's interest and support.

1. In the introduction, general statements are made regarding axial proteins recruiting HORMA proteins through closure motifs in yeast, mammals and plants. I don't think that this has been shown conclusively in all cases. For example, as far as I'm aware, the only evidence regarding SYCP2 is a Y2H interaction between a long fragment and HORMAD2. I think some caveats should be added, and ideally specific references of what has been shown, to ensure that readers are aware of the uncertainties.

*We have added the caveats to the introduction that the recruitment mechanisms are not fully explored. However, an interaction between SYCP2 and HORMAD2 using purified proteins in a pulldown, and recombinant HORMAD2 with a SYCP2 peptide in fluorescence polarisation has also been shown by the Corbett laboratory (West *et al.*, 2019, *eLife*, Figure 2, Supplement 2), in addition to their Y2H data in the main figure.*

2. Figure 2C - The overall AF2 model is likely to be correct given the similarity to other HORMA proteins. Why did the authors choose CM-1 rather than CM-2 as the bound closure motif? It would be helpful to see details of the side-chain interactions at the closure motif interface (in the model and in the experimental structure), along with an alignment, to see whether they appear plausible. Also, what do the authors make of the N-helices in the AF2 models? Do the sequences look as though they should be helical, are they largely hydrophilic or do they have a potential heptad repeat that could imply involvement in a coiled-coil structure?

We chose one CM in the self-bound form for what is now Figure 1F, but the two predicted structures are similar. We have completed a more detailed analysis of CM-1 and CM-2 (now referred to as CM-H1-A and CM-H1-B to avoid confusion with the non-functional but predicted CM in ecHOP1-isoform 2). We have not made a direct comparison with other CM-HORMA interactions, since the motif is very degenerate. This analysis is in Figure EV3 (for CM-H1-A vs. CM-H1-B) and EV4 for CM-H1-A. Both closure motifs share key similarities in their predicted binding mechanisms, and both are plausible. This has also formed the basis for the comparison with the ecRED1 closure motifs (summarised in Figure 4F).

The reviewer raises an interesting point with the alpha-N helix of ecHOP1. The alpha-N helix is not especially hydrophobic, but only has a weak heptad repeat. Based on this, and on the fact that ecHOP1 predominantly forms monomers we would not, at this stage, speculate that it is a coiled coil. Constructs of ecHOP1 containing the alpha-N helix are monomeric (see the response to the reviewer's point 4 below).

3. Figure 3B - Whilst I am confident with the interpretation that the protein is a monomer, the MW is stated with false precision at 34.09 kDa in the figure and 34.01 kDa in the main text. The MW fit across the curve shows deviate that far exceeds the stated precision, and other limitations mean that MALS data are typically accurate to between 5-10 kDa (i.e. more than sufficient to tell whether it is a monomer or dimer etc). I recognise that an error range is shown, but such false precision may mislead readers who are unfamiliar with the technique. I would suggest that the MW should be stated as approximately 34 kDa. Also, it would be helpful if it were plotted on a linear scale so the range of

MWs across the peak could be interpreted. Further, the theoretical MW of a monomer should be stated in the figure legend.

We acknowledge the concern about overly precise MW values. We will revise both the figure and text to report the MW as "approximately 34 kDa," and include the theoretical MW of a monomer in the figure legend. We will reformat the SEC-MALS plot to a linear scale, clarifying the range of MWs across the peak.

4. Figure 3B and S3 - I am not convinced that the 57 kDa species is a real second species as the absorbance barely shows a peak above the baseline, so may simply represent noise. Did the authors try analysing material at different concentrations to see whether there is concentration-dependent assembly into a dimer? If they want to include this, they should integrate the two peaks separately and state the percentage of material by mass in each peak so the relative amount is clear to readers (I imagine the second peak will be <1%). I think the AF2 modelling of dimers is very speculative as AF2 comes up with the best model assuming that the given oligomer is correct, and I have seen similar and far higher levels of confidence for completely spurious associations. It is good that they performed the pull-downs upon deleting the N-helices. On this basis of the data shown, I would say after investigating whether it could form a dimer, they have determined that it seemingly remains monomeric. I suggest editing the text to prevent readers from misinterpreting these points (e.g. I would remove the Af2 dimer modelling and simply state that they tested whether it could form a dimer with/without the N-helix and found no evidence for self-association).

We fully appreciate the reviewer's caution here. Current evidence for echOP1 dimerization is weak, and the 57 kDa signal might indeed represent background noise. We carried out SEC-MALS at higher concentrations and did not observe an increase in the 57 kDa peak. We concede that the discussion on echOP1 dimerization could cause confusion and we have elected to remove this from the manuscript.

5. Figure 3C-E,K - I would suggest some caution in interpreting the AF2 models as it tends to hallucinate interfaces for well-known interacting proteins, meaning that most conserved regions of unstructured sequence are predicted to bind as closure motifs when modelled together. The key discriminator is whether the side-chain interactions are favourable. Hence, I suggest showing panels of the predicted side-chain interfaces of the putative CMs in their complexes in comparison with the experimentally determined interfaces (that could be shown in figure 2). The authors should discuss in the text whether the predicted interfaces appear plausible. In relation to this, it is good that the authors tested these interactions in pull-downs (with targeted mutations) and by ITC. It is notable that CM1-2 had a confident AF2 complex but failed to show binding by ITC - might this be an example of such an hallucination?

This comment accurately captures a key focus of our study. We fully agree that AlphaFold2 models, while powerful for hypothesis generation, can indeed generate incorrect interfaces. Thus, we have systematically used experimental approaches (pull-down assays and ITC) to validate predicted interactions rigorously.

In Figure EV3 we provide a detailed comparison of the binding of CM-H1-A and CM-H1-B and their conservation. Both are conserved, and use similar binding interfaces to one another. Similarly, CM-H2-A (that did not bind to echOP1-HORMA) has an interface that lacked many of the contacts shown for CM-H1-A, CM-H1-B and the ecRED1 closure motifs (summarised in Figure 4F).

6. Figure 3K - where is the evidence for this trans-interaction given that CM1-2 failed to bind to the HORMA domain? As above, I would only present AF2 models if they are supported by experimental evidence.

*We thank the reviewer for this point. Indeed CM1-2 (now referred to as CM-H2-A) does *not* bind to the HORMA domain. However Figure 3K shows CM2-1 (now CM-H1-B) (and CM1-1; now CM-H1-A) bound to the HORMA domain, which is the second predicted CM in isoform 1 of echOP1, and does bind to the HORMA domain.*

In re-reading the manuscript we realise that this confusion has probably arisen due to our chosen nomenclature for the closure motifs in the HOP1 isoforms, which is not ideal. We have thus changed the terminology for the closure motifs to CM-H1-A (closure motif - HOP1 - isoform 1 - A), CM-H1-B (closure motif - HOP1 - isoform 1 - B), CM-H2-A (closure motif - HOP1 - isoform 2 - A) and CM-R-A to CM-R-F for the putative ecRED1 closure motifs.

7. Figure 3J - This is a pedantic point, but it is incorrect to say that $K_d=0$, as it is not determined to be zero by the assay. Instead, ITC performed using these protein concentrations with associated limitations etc failed to detect an interaction, and the experimental setup would not detect a low affinity interaction (e.g. $> 1\text{mM}$). Instead, I would state $K_d=n.d.$ (not determined).

The reviewer is correct, and we will change to $K_d=n.d.$

8. Line number 291 - SYCP2 has not been found to form homotetramers. SYCP3 forms mostly homotetramers and may also form 2:2 complexes with SYCP2.

The reviewer is correct, and we have amended accordingly (lines 382-384)

9. Figure S5 - Has ecRED1 been shown to form a tetramer? Has the AF2 model been validated experimentally, and do the side-chain interactions appear plausible? Models of coiled-coils are very often incorrect even when there is high confidence, so I would only show such models if there is appropriate validation.

We have spent some time on recombinant approaches to the ecRED1 coiled-coil region to address this question. The full coiled-coil region showed the formation of high-order assemblies, but negative stain EM could not resolve this assemblies as previously reported filaments. Based on previous work (Corbett and Davies laboratories) we produced a ecRED1-coiled-coil ΔC_{tip} construct that should have prevented assembly formation. This construct behaved well during purification, but to our surprise provided a measured mass of 435 kDa (see below) which would have been between a hexamer and heptamer. We reasoned that this was unlikely, and found that we had in fact a beautiful purification of GroEL. In the interests of time we have paused work on the ecRED1 coiled-coil and left the speculation in place that it may form tetramers, which may in-turn form filaments.

10. Figure 5E - As previously, side-chain interactions of the putative closure motif should be shown to determine whether they appear realistic.

We fully agree. We will have added LigPlot representations of side-chain interactions for CM-H1-A and CM-H1-B (Figure EV3). We have also shown a detailed 3D representation of CM-H1-A binding to HORMA (Figure EV4) and summarised the relevant side-chain interactions for all CMs in Figure 4F.

Referee #3:

General Summary

This manuscript explores how meiotic axis formation is regulated in the model brown alga *Ectocarpus*, focusing on two proteins predicted to function analogously to the well-characterized Hop1 and Red1 in other eukaryotes:

1. ecHOP1: A HORMA-domain protein with two isoforms. Structural and biochemical data reveal that ecHOP1 harbors a winged helix-turn-helix (wHTH) domain that binds dsDNA, and up to three predicted closure motifs (CMs), two of which (CM1-1 and CM2-1) are shown to bind in cis to its own HORMA domain.

2. ecRED1: Contains several predicted CMs (the best validated is CM1) capable of binding the ecHOP1 HORMA domain. This suggests ecRED1 can recruit ecHOP1, likely assembling the chromosome axis during meiosis.

In addition to biochemical pull-downs (including mutated motifs), ITC, and SEC-MALS, the authors present transcriptomic data revealing that these genes are selectively expressed in sporophyte meiotic tissues but are absent in parthenosporophytes undergoing a form of non-reductive (apo)meiosis. These findings position *Ectocarpus* as a strategically important model for studying meiotic diversity, expanding our evolutionary understanding beyond classical plant, animal, and fungal systems.

*We thank Reviewer #3 for their detailed, thoughtful feedback and for recognizing the evolutionary significance and broader implications of our study. We especially appreciate their acknowledgment of *Ectocarpus* as an important model organism to broaden our understanding of meiotic diversity across eukaryotes.*

Opinion & Principal Significance

A. Mechanistic Insight into Meiotic Axis Assembly: The discovery that ecHOP1 has multiple closure motifs and a robust dsDNA-binding domain highlights a flexible mechanism for chromosome binding. This flexibility could regulate break formation or checkpoint signaling in ways distinct from yeast or vertebrates.

B. Evolutionary Perspective: Ectocarpus diverged from canonical models; nonetheless, ecHOP1/ecRED1 share fundamental HORMA and Red1-like features. This supports the idea that meiosis evolved only once (in the LECA) but diversified via lineage-specific rearrangements (e.g., additional/alternative motifs).

C. Comparative Genomics: By examining ecHOP1 and ecRED1 alongside known homologs (Hop1, Red1, SYCP2, HORMAD1/2), the manuscript broadens our phylogenetic understanding of meiotic proteins. It paves the way for future comparative studies on additional Stramenopiles.

D. Study Limitations: while the biochemical data and AlphaFold predictions are compelling, in vivo localization or functional assays (e.g., immunofluorescence, gene tagging) are still lacking. Developing such tools in Ectocarpus would further strengthen the conclusions on ecHOP1's recruitment, post-translational modifications, and overall axis architecture.

Points of Attention

A. Functional Validation of Predicted Closure Motifs - The authors have tested all predicted CMs in ecHOP1. They show that CM1-1 and CM2-1 indeed bind the HORMA domain, whereas CM1-2 does not. Thus, for ecHOP1, these motifs appear fully explored. By contrast, ecRED1 has multiple CMs predicted by AlphaFold, but only CM1 (and perhaps weak binding by CM4) is clearly demonstrated to interact with ecHOP1 in vitro. The other predicted sites remain unverified. It would strengthen the paper to clarify why these putative CMs did not show binding (e.g., tested but negative? conditionally active?). Also could the authors show the AF2/AF3 data for these CMs and summarise their iPTM values and any other confidence values to give the reader an indication of how relevant each of the CMs is? In general it is not entirely clear to me what the exact reasoning is to classify each CM beyond maybe a conserved arginine/lysine. Could the authors expand on this and make it more complete? Suggestion is to expand the discussion to explain whether the unvalidated ecRED1 motifs are false positives, require post-translational modifications, or might be engaged under specific biological conditions. In case of PTMs these might be tested using AF3?

We thank the reviewer for this point. Indeed we have thoroughly tested all candidate CMs for ecHOP1 in both in vitro pull-down and in ITC. We considered this necessary because pulldowns suggested that all three ecHOP1 CMs could bind to the HORMA domain, but ITC showed that CM1-2 did not bind, and that the pulldown was likely an artefact (already suggested due to the mutant CM still binding in the pulldown). For ecRED1 CMs we tested all candidate sequences in pulldowns, but only CM1 showed strong binding.

Figure 4B shows the iPTM score summary of the top 5 iPTM scores for 25 predictions for each of the ecRED1 CMs. In Figure 4F we summarise simply the side chain interactions of the ecRED1 CMs, and compare these to the ecHOP1 CMs. We find that, purely based on the structural prediction, several ecRED1 CMs are plausible, and CM-R-D seems to be even better than the other motifs.

We have used predictive tools to explore phosphorylation and SUMOylation. We are challenged with the phosphorylation in that the consensus kinase sequences and phosphoproteome in Ectocarpus remain unexplored. Different predictive tools suggest that a large number of serines and threonines could be phosphorylated in ecRED1 and ecHOP1. For SUMOylation sites and SUMO interaction motifs (SIMs) there are fewer predictive sites. However, AlphaFold modelling does not support the location of the SIMs in ecHOP1 or ecRED1, when using either the putative Ectocarpus SUMO or mammalian SUMO sequences. Thus we have decided to limit the speculation in the manuscript to a few lines and one supplementary figure.

B. Temporal Regulation and Post-Translational Modifications - Yeast Hop1 and mammalian HORMADs are regulated by phosphorylation or SUMOylation to tune break formation and checkpoint responses. The data here indicate that ecRED1 and ecHOP1 might also be modified, yet no direct evidence (phospho-mutants, MS data) is provided. Even a short pilot experiment or extended discussion of potential kinase/ATPase homologs (Mek1, Pch2) would help the reader appreciate the possible regulatory circuitry.

We thank the reviewer for this suggestion. Identification of *Mek1* orthologs outside of budding yeast is challenging. Multiple kinases in *Ectocarpus* show the same FHA-Kinase domain organisation of *S. cerevisiae* *Mek1*. For *Pch2/TRIP13* however we did identify a more probable ortholog (*Ec-19_002220*). AlphaFold modelling suggested that this factor had a similar fold to *Pch2/TRIP13*. When we modelled this with *ecHOP1*, AlphaFold did not predict an interaction (see below). Due to the inconclusive nature of these findings, we did not include this in the resubmission.

C. Alternative (Apo)Meiosis in *Ectocarpus* - The authors note that *ecHOP1/ecRED1* expression is absent in parthenosporophytes, which undergo a non-reductive (apo)meiosis. This difference is intriguing for understanding how the chromosome axis might diverge in non-canonical meiosis. The manuscript would benefit from acknowledging whether other axis-like factors could be present in pSP stages, or if those algae bypass the need for a classical axis. Even speculation would clarify broader evolutionary ramifications.

We thank the reviewer for highlighting this intriguing aspect. Our RNAseq data indeed show *ecHOP1* and *ecRED1* expression only during canonical meiosis, suggesting that parthenosporophytes (pSP) either bypass classical axis formation entirely or utilize alternative axis-like factors not yet identified. We have added a sentence to this effect (lines 160 to 163).

Minor Concerns & Editorial Points

A. Protein Nomenclature & Consistency

a. Standardize *ecHOP1* vs. *EcHop1*, etc.

b. Clarify your notation for isoforms (e.g., "*ecHOP1*^{Iso1}," "*ecHOP1*^{Iso2}," or "*ecHOP1*^{HORMA¹-274}") to ensure clarity when discussing distinct constructs.

Thank you, we will clarify this.

B. RNA Expression Analysis - Briefly mention whether you used housekeeping genes or how you handled normalization. Also clarify the statistical tests (e.g., T-tests) for differential expression.

*As requested, we will briefly note that we used TPM values (\log_2 -transformed) for normalization and performed pairwise *t*-tests for each tissue comparison. Although housekeeping genes were not used, TPM is a standard approach to robustly compare expression across samples.*

C. Contextual/Line-Specific Items

- Line 46-49: Strengthen the justification for using a stramenopile model and how that addresses "large evolutionary context."
- Line 128-136: Reads more like an introduction than results. Consider relocating.
- Minor Typos:
 - o Line 55: remove "the" before "gametophyte."
 - o Line 195-200: check for doubled sentences.
 - o Line 344: missing comma after "motif."
 - o Line 620: "lised" → "lysed."
 - o Line 971: "SCYP2" → "SYCP2."
- ORCAE: Define the acronym or website context upon first mention.

We accept these suggestions and have strengthened the rationale for choosing Ectocarpus as a representative Stramenopile model, emphasizing how it addresses evolutionary diversity of meiosis mechanisms. We have also relocated and condensed introductory-style material from Results into the Introduction, maintaining logical manuscript flow.

We also appreciate and accept all corrections indicated.

D. Hop1-Related Points

- Figure 2E (percent identity/similarity) may be less informative than a proper phylogeny showing echHOP1 clustering with known Hop1 orthologs (and an outgroup of another HORMA protein).

We have added a phylogeny for echHOP1 vs. known Hop1 orthologs (Figure 1G)

- If two isoforms exist in Ectocarpus, clarify whether other brown algae also share these predicted isoforms, suggesting deeper evolutionary conservation.

This is a good point, however there is currently limited experimental data available from other brown algal species for us to be able to properly address this point.

- In Supplementary Figure 1B, highlight which species have the extra β -sheet region ($\beta 3'/\beta 3''$) - include representatives of other eukaryotic clades. If it's truly unique to brown algae, a brief speculation on function beyond other interactors would be valuable (perhaps an AF-based interactor screen would be an option?).

We thank the reviewer for this suggestion. We are in the process of looking for interactors - both novel and established - for echHOP1. To this end we have undertaken both a yeast-2-hybrid screen, and an AlphaFold based screen, which we will address in detail in the future.

E. wHTH Domain

- The multi-sequence alignment (Figure 4G) might be more approachable if limited to Stramenopiles (alignment seems not very strong with only three very distant organisms in it). If referencing widely divergent taxa, clarify their potential structural differences (single vs. double wHTH domains - which of the two domains is used?).

We have limited the wHTH MSA to Stramenopiles, which is now in the modified Figure 3, and as a new supplementary figure (Figure EV6). The majority of the Stramenopiles contain a single wHTH, however we found two species that contain two wHTH domains. These are highlighted in EV6.

- Consider performing an AF3 run with ecHOP1 bound to dsDNA to refine the interaction model (any clashes or additional insights?).

We have performed these experiments, however, AF3 frequently produces poor quality DNA-protein interaction predictions, even with well established DNA binding factors. In the case of ecHOP1-wHTH, we tried with many different DNA sequences (including those sequences we used in wet lab experiments) but binding was only intermittently predicted.

F. Alignments

Throughout the manuscript various multiple alignments are shown. It would be good to use a consistent coloring scheme and perhaps also a consistent tool to generate the alignments. My advice is to the common jalview scheme that many are used to - but this is up to the authors of course. I am mentioning jalview as the use of such tools make sure all residues that are supposed to be in the same column are properly aligned - now in suppl figure on the CMs in ecRed1 this is not the case. Also the coloring scheme here doesn't help to assess how conserved positions really are. This would help this type of analysis considerably as I think this is an excellent figure in general. Sidenote - did the authors consider adding in disorder prediction to assess whether these CMs are in fact available to bind to ecHop1 and are not in a globular region?

We have used a consistent coloring scheme and presentation style across all alignments. We have also added a disorder prediction (AIUPred) to the domain cartoons for ecHOP1 and ecRED1 which does show that the CMs are in disordered regions

Additional Suggestions for Improving the Study

A. In Vivo Localization Studies - while challenging, immunofluorescence or gene tagging would directly confirm how and where ecHOP1 and ecRED1 localize in meiotic cells.

We fully agree with the reviewer that this type of approach would be insightful. We have already raised two rounds of antibodies against ecHOP1, and neither of these were suitable for IF work. Regarding tagging, this is currently not technically possible in Ectocarpus.

B. Expanded DNA-Binding Specificity - assessing whether ecHOP1 prefers certain DNA structures (e.g., supercoiled, specific sequences, nucleosomal) could illuminate how break hotspots might be designated in brown algae.

We have added EMSAs looking at whether the wHTH of ecHOP1 binds preferentially to different DNA shapes (Figure 3G), and to different lengths of dsDNA (Figure 3F). Other than the clear preference for dsDNA over ssDNA we note no other clear preference for ecHOP1 DNA binding.

C. Post-Translational Regulation - since PTMs are pivotal in many meiotic systems, systematically probing for phosphorylated or SUMOylated residues (and their functional consequences) would be a logical next step (some of it might be show already with AF3).

We agree that proteomics work from meiotic algal tissue would be very interesting, however we would argue that such large-scale experiments are somewhat beyond the scope of this study. As described above we have included a brief discussion on the potential role of phosphorylation and SUMOylation.

Overall Recommendation

This manuscript provides novel insights into how a meiotic axis may be assembled in a non-traditional eukaryotic model and lays the groundwork for future comparative approaches. The authors have tested all predicted CMs in ecHOP1 (clarifying which ones do or do not bind), but several ecRED1-predicted motifs remain unconfirmed. With some additional detail on unvalidated CMs, a deeper discussion of possible post-translational modifications, and some future follow-up in vivo confirmation, this study will represent a valuable contribution to our understanding of meiosis across the eukaryotic tree.

We thank the reviewer for their supportive comments, and their detailed review. We believe we can address the vast majority of their suggestions, which will strengthen the manuscript overall.

Referee #4:

The manuscript "Characterization of meiotic axis proteins in the model brown alga *Ectocarpus*" by Kane et al. describes the first characterization of the meiotic axis proteins HOP1 and RED1 in the brown alga *Ectocarpus*, an emerging model system for the study of meiosis. HOP1 is a member of the broad HORMA domain protein family of meiotic recombination regulators, and RED1 is a so-called "axis core" protein putatively involved in organizing meiotic chromosomes along with meiosis-specific cohesin complexes. Briefly, the authors demonstrate that HOP1 and RED1 are expressed specifically in the meiotic life cycle stage; identify two isoforms of HOP1 and dissect their domain architectures; dissect the "closure motifs" in both HOP1 and RED1 that bind the HOP1 HORMA domain, and show that the HOP1 winged helix-turn-helix domain binds DNA.

Overall, there is interesting data here, but I don't feel that there is sufficient novelty to warrant publication in *The EMBO Journal*. The novel features of these proteins, including their putative possession of multiple closure motifs, mainly rest on inconclusive AlphaFold predictions, and are not sufficiently justified or supported by their experimental data.

We thank the reviewer for their comment. We have indeed used AF2/3 as a "fishing" tool, to find potential CMs. As reviewer 2 points out, the algorithm can "hallucinate", this is why we have taken the time to explore each of the potential CMs using biochemical and biophysical approaches. We apologise that we were not clear enough in the manuscript that we are only proposing that the experimentally validated closure motifs, are relevant closure motifs. These are the two closure motifs within the ecHOP1 isoform1 sequence, and the CM1 of ecRED1.

We have speculated in the manuscript that, in the case of the predicted ecRED1 CMs, those that were not experimentally validated could become relevant under certain conditions including, for example, the addition of post-translational modifications. We have left this speculation in the resubmission, since the predicted side-chain interactions of CM-R-B to CM-R-F are consistent with what has been shown for experimentally validated CMs. Therefore it is not completely unreasonable to propose that these cryptic CMs might become functional under certain conditions.

Specific points:

The authors should make some attempt to discern which isoform of ecHOP1 is more prevalent. Does their PCR indicate that isoform 1 is more prevalent? Is there evidence of which ecHOP1 isoform predominates from the RNA-Seq dataset the authors draw from in Figure 1?

We agree that determining the relative abundance of each ecHOP1 isoform is of interest. From our RNA-seq data (the same dataset used in Figure 1), we find that isoform 1 is slightly more abundant overall, though both isoforms are expressed at relatively low levels. We will include a brief mention of this observation in the revised manuscript to clarify that, while isoform 1 appears somewhat more prevalent, both isoforms are indeed present.

Our rtPCR assays also detected both isoforms, though the faintness of the second band likely reflects the lower expression level of isoform 2. We plan to provide an adjusted image or in Figure 2B to better illustrate amplification from both isoforms.

Related to the above, in Figure 2B (Gel 3), the 276 bp band is difficult to visualize. A darker exposure or adjusted contrast may help clearly show amplification from both ecHOP1 isoforms.

We will correct this.

I don't see the gray arrow in figure 2C as stated in the text.

Thank you for pointing this out; this arrow should now refer to Figure 1F.

Lines 157-183: The authors seem to have deleted any description of how the different putative closure motifs in the two ecHOP1 isoforms were identified from this section. Did they perform AlphaFold with both isoforms? Did the closure motifs show confident interactions with the HORMA domain? I know some of these questions are addressed later, but it's jarring that they are not addressed in this section, especially given the phrasing of the last sentence in this section.

We thank the reviewer for this observation. We will update the manuscript to show that all three putative ecHOP1 closure motifs were predicted by AlphaFold2, and display the confidence scores as for ecRED1 CMs. This should further serve to emphasise the need to experimentally validate the CMs, as the reviewer has pointed out, since all three of the ecHOP1 CMs were predicted with equivalent confidence.

Supplementary figure 3: please report AlphaFold ipTM (interface pTM) values, which report on the confidence of complex formation. (Currently pTM scores are noted, which is not the same).

Thank you, we have changed this.

Also related to Supplementary Figure 3: The HOP1 dimer models shown in panels B and D are not consistent with one another. Given this inconsistency, and the authors' failure to confirm HOP1 dimerization experimentally, I suggest leaving out this inconclusive data.

This was also raised by reviewer 2. In line with this reviewer's suggestion, and the weak data, we have removed all speculation on ecHOP1 dimer formation from the manuscript.

For the three ecHOP1 closure motifs identified by the authors, does the sequence alignment shown in Figure 3F match up with the register predicted by AlphaFold for these motifs' binding to the ecHOP1 HORMA domain?

We confirm that the alignment in Figure 3F is designed to reflect the same register as that predicted by AF2 for each motif's interaction with the HORMA domain.

Line 301: reference to figure 5D should be Figure 5B.

We thank the reviewer for catching this. We will correct the figure reference from Figure 5D to Figure 5B.

Where is the evidence that AlphaFold 2 identified six putative closure motifs in ecRED1? There should be supplemental figure panels showing more detailed AlphaFold results supporting Figure 5B.

We have shown example detailed interactions for CM-H1-A and CM-H1-B (Figure EV3 and EV4). We have then used Figure 4F to summarise the key interactions between the CMs and the ecHOP1-HORMA. In this manner, we hope to convey the idea that, while CM-H2-A is clearly a poor candidate for a CM, it was not possible to clearly differentiate the one experimentally validated ecRED1 CM (CM-R-A) from the remaining five.

More broadly, if the authors' data indicate that there is only one functional closure motif in ecRED1 (which it does), why even mention the other five? This gets to a larger issue running throughout the paper: describing inconclusive/low confidence AlphaFold predictions, then showing that experiments cannot confirm these predictions.

Given the possibility that AlphaFold is simply wrong, why are these predictions reported if they can't be supported with experimental evidence?

This important comment highlights a key conceptual point in our manuscript. Our rationale for including multiple predicted motifs was to illustrate the strengths and limitations of AlphaFold as a predictive tool, emphasizing that experimental validation - especially thorough biochemical work - remains essential. We have endeavoured to communicate this concept more effectively in this resubmission. We also acknowledge that undifferentiated highlighting of motifs, particularly in main figures, which lack experimental validation might cause confusion. We have clearly distinguished experimentally validated motifs from predictions that remain speculative, using stronger colours for CM-H1-A, CM-H1-B and CM-R-A.

Dear Dr. Weir

Thank you for the submission of your revised manuscript to EMBO reports. I apologize for the delay in handling your manuscript but we have meanwhile received the full set of referee reports that is copied below.

As you will see, all referees are very positive about the study and support publication after a few remaining concerns regarding closure motifs and figures have been addressed.

From the editorial side, there are also a few things that we need before we can proceed with the official acceptance of your study.

- Your manuscript will be published in our Reports section and therefore needs a combined Results and Discussion section. The character count should be roundabout 27,000 characters (including spaces but excluding materials & methods and references).

- The manuscript sections should be in the following order: Title page - Abstract & Keywords - Introduction - Results - Discussion - Methods - Data Availability - Acknowledgments - Disclosure Statement & Competing Interests - References - Figure Legends - (Main Tables with legends if applicable) - Expanded View Figure Legends.

- Materials and Methods should be Methods.

- "Extended View Figures and Legends" should be "Expanded View Figure Legends".

- Please remove the figures from the manuscript text file. Only the legends of main and EV figures remain and these should be placed at the end of the manuscript, after the References.

- Figure callouts: Supplementary Table 1 and 2 are called out but missing. Please add callouts for Figure 4E. Supplementary figure 1A is a wrong callout. Should it be Figure EV1A?

- Please reduce the number of keywords to 5.

- The conflict of interest statement needs the header "Disclosure and Competing Interests Statement".

- Please remove the ORCID IDs from the manuscript file. It is sufficient to have them in the manuscript tracking system.

- Please provide a complete author checklist, which you can download from our author guidelines (<<https://www.embopress.org/page/journal/14693178/authorguide>>). Please insert information in the checklist that is also reflected in the manuscript. The completed author checklist will also be part of the RPF.

- Please provide a Reagents and Tools Table listing key reagents, experimental models, software and relevant equipment and including their sources and relevant identifiers. You can download the template (.docx), from our author guidelines: <https://www.embopress.org/page/journal/14693178/authorguide#structuredmethods>.

- You currently have 9 EV figures. We used to have a limit of 5, but are more flexible now. That said, you might want to reduce them a bit, e.g., by combining some that have fewer panels?

- We perform a routine image integrity check on all revised manuscript and noticed the following points that need to be clarified:
a) The blots shown in Figure 2A are reused in Figure EV2. Please clarify and clearly state the reuse in the respective figure legends.

- b) The blots shown in Figure 3I have been reused in Figure EV6 D&E. Again, please clarify and clearly state the duplicate use in the respective figure legends.

- Please deposit the RNAseq dataset in a public repository and provide the accession numbers and the link in a dedicated Data Availability section at the end of the Methods. Suggested wording: "The [structural coordinates | microarray | mass spectrometry] data from this publication have been deposited to the [name of the database] database [URL] and assigned the identifier [accession | permalink | hashtag].". Please note that the link needs to resolve directly to the dataset, not just the database.

- Evans et al (2021) is a preprint. Please cite it as (preprint: Evans et al, 2021) in the text and add [PREPRINT] at the end of the reference in the reference list.

- Lipinska et al, 2017 and Lotharukpong et al, 2024 can be turned into data citations. If you want to do so, you would need to add the URL that links to the dataset you have used to the reference list (with the same authors). This is marked with [DATASET] at the end of the reference. In the text you would cite it as (Lipinska et al, 2017; Data ref: Lipinska et al, 2017). The first reference is to the paper that generated the dataset and the second reference is that of the dataset itself. See also <<https://www.embopress.org/page/journal/14693178/authorguide#referencesformat>>.

- Our production/data editors have asked you to clarify several points in the figure legends (see below). Please incorporate these changes in the manuscript and return the revised file with tracked changes with your final manuscript submission.

- Please provide a scale bar for figure 1B.

- Please indicate the statistical test used for data analysis in the legend of figure 1C.

- Please note that the box plots need to be defined in terms of minima, maxima, centre, bounds of box and whiskers, and percentile in the legends of figure 1C.

- Please note that information related to n is missing in the legends of figure 1C.

- Please note that the error bars are not defined in the legends of figure 3C.

- Finally, EMBO Reports papers are accompanied online by

A) a short (1-2 sentences) summary of the findings and their significance,

B) 2-3 bullet points highlighting key results and

C) a schematic summary figure that provides a sketch of the major findings (not a data image).

Please provide the summary figure as a separate file in PNG or JPG format at a size of 550x300-600 pixels (width x height).

Please note that the size is rather small and that text needs to be readable at the final size. Please send us this information along with the revised manuscript.

With kind regards,

=====

Referee #1:

The authors have thoroughly addressed all of my previous comments and I agree entirely with all of their points and conclusions. Also, I thank the authors for highlighting one bit of data (relating to SYCP2-HORMAD2) that I had missed from a supplementary figure of a published paper, which is relevant to their introduction. They have done an excellent job of revising the manuscript, which I think has benefited nicely from the round of review. I am happy with all of the changes, and am delighted to recommend that the current submission is accepted for publication in EMBO Reports.

Referee #2:

Upon reading the revised version of the manuscript and the rebuttal, I conclude that the authors have gone the distance to alleviate many of the issues/concerns I have raised in my previous review. Although their findings might not be wholly novel, the work is well executed and is important for two reasons:

(1) it shows how comparative genomics approaches can inform the selection of eukaryotic organisms for further biochemical (and hopefully also cell biological) exploration of meiotic systems. In turn this will open up the much needed discussion in the meiotic field on how non-model eukaryotes execute meiosis. In this sense I laud the efforts of the authors.

(2) this body of work combines biochemistry with alphafold predictions for a non-model eukaryotes for which cell biological experiments are yet inaccessible. As such the authors provide a blue-print/template for others on how to integrate biochemistry and alphafold prediction into their work. Especially alphafold-type predictions now give us a wide array of hypotheses to test and this manuscript illustrates that simply focussing on conserved features only will not give you the full picture (multiple cryptic CMs are involved in Hop1-Red1 binding).

A last note for the authors - it would have been good to see a bit more extended evolutionary/comparative insights to better contextualise their findings, but the current analyses hold and are informative to support experimental choices making this an excellent manuscript to be published in EMBO reports.

Referee #3:

In this resubmission, the authors have responded comprehensively to most concerns and questions by the three reviewers. I remain unsatisfied, however, with several aspects of their presentation of putative versus experimentally validated closure motifs in ecHop1 and ecRed1. I don't think the authors should call something a "closure motif" simply because there is (extremely) weak sequence homology to a known closure motif (as in CM-H2-A) or if AlphaFold creates a model with a somewhat believable ipTM (as in CM-R-B through CM-R-F).

The authors seemingly ignored my specific concern that AlphaFold data (model and PAE plots) for Hop1 binding CM-R-B through CM-R-F is lacking - simply showing the ipTM ranges in Figure 4B is not enough (especially given that most of these ipTMs are well below the range for a confident prediction given by the AlphaFold developers: 0.6-0.8 is considered marginal confidence, and anything below that is likely not a valid interaction). This data needs to be added.

Given that CM-H2-A and CM-R-B though CM-R-F have been tested biochemically and show no interaction, I feel strongly that these should not be labeled as closure motifs at all. They are simply candidate sequences that AlphaFold or sequence alignments predicted, which the authors (to their credit) have convincingly shown are NOT closure motifs. It is therefore misleading to call them closure motifs. The authors' justification that their goal in presenting the various spurious predictions is "to illustrate the strengths and limitations of AlphaFold as a predictive tool, emphasizing that experimental validation - especially thorough biochemical work - remains essential" is not satisfactory.

I don't think major changes are needed to address this concern. Minor figure alterations to present candidate sequences simply labelled by their protein and residue range (not as "closure motif X"), along with adding the models and PAE plots for ecRed1 candidates as mentioned above, should be sufficient. To be clear, I think the authors are being rigorous, and they are clear in the text about what has, and has not, been experimentally validated. My main concern is that a casual reader will draw the wrong conclusions from the figures, for example seeing Figure 4A and Figure 5 and thinking "Ah, ecRed1 has six closure motifs - just like *C. elegans* HTP-3!" when this is emphatically NOT supported by the data.

Specific points:

Figure 4F: Legend appears to be missing, and more importantly it's not very clear what the authors are trying to convey here. To be clear, I understand what they are trying to convey - it's just not obvious from the design of the figure and lack of an accompanying visual aid.

Figure EV8: It is impossible to judge from this figure whether these putative closure motifs in brown algae Red1 proteins are conserved relative to the surrounding (presumably disordered and poorly conserved) protein regions or not.

Rebuttal to Resubmission August 2025

Referee #1:

The authors have thoroughly addressed all of my previous comments and I agree entirely with all of their points and conclusions. Also, I thank the authors for highlighting one bit of data (relating to SYCP2-HORMAD2) that I had missed from a supplementary figure of a published paper, which is relevant to their introduction. They have done an excellent job of revising the manuscript, which I think has benefited nicely from the round of review. I am happy with all of the changes, and am delighted to recommend that the current submission is accepted for publication in EMBO Reports.

We thank Reviewer 1 for their constructive comments and their support in publication in EMBO reports.

Referee #2:

Upon reading the revised version of the manuscript and the rebuttal, I conclude that the authors have gone the distance to alleviate many of the issues/concerns I have raised in my previous review. Although their findings might not be wholly novel, the work is well executed and is important for two reasons: (1) it shows how comparative genomics approaches can inform the selection of eukaryotic organisms for further biochemical (and hopefully also cell biological) exploration of meiotic systems. In turn this will open up the much needed discussion in the meiotic field on how non-model eukaryotes execute meiosis. In this sense I laud the efforts of the authors.

(2) this body of work combines biochemistry with alphafold predictions for a non-model eukaryotes for which cell biological experiments are yet inaccessible. As such the authors provide a blue-print/template for others on how to integrate biochemistry and alphafold prediction into their work. Especially alphafold-type predictions now give us a wide array of hypotheses to test and this manuscript illustrates that simply focussing on conserved features only will not give you the full picture (multiple cryptic CMs are involved in Hop1-Red1 binding).

A last note for the authors - it would have been good to see a bit more extended evolutionary/comparative insights to better contextualise their findings, but the current analyses hold and are informative to support experimental choices making this an excellent manuscript to be published in EMBO reports.

We thank Reviewer 2 for their support. We look forward to providing more evolutionary/comparative insights in the future.

Referee #3:

In this resubmission, the authors have responded comprehensively to most concerns and questions by the three reviewers. I remain unsatisfied, however, with several aspects of their presentation of putative versus experimentally validated closure motifs in ecHop1 and ecRed1. I don't think the authors should call something a "closure motif" simply because there is (extremely) weak sequence homology to a known closure motif (as in CM-H2-A) or if AlphaFold creates a model with a somewhat believable ipTM (as in CM-R-B through CM-R-F).

The authors seemingly ignored my specific concern that AlphaFold data (model and PAE plots) for Hop1 binding CM-R-B through CM-R-F is lacking - simply showing the ipTM ranges in Figure 4B is not enough (especially given that most of these ipTMs are well below the range for a confident prediction given by the AlphaFold developers: 0.6-0.8 is considered marginal confidence, and anything below that is likely not a valid interaction). This data needs to be added.

Given that CM-H2-A and CM-R-B though CM-R-F have been tested biochemically and show no interaction, I feel strongly that these should not be labeled as closure motifs at all. They are simply candidate sequences that AlphaFold or sequence alignments predicted, which the authors (to their credit) have convincingly shown are NOT closure motifs. It is therefore misleading to call them closure motifs. The authors' justification that their goal in presenting the various spurious predictions is "to illustrate the strengths and limitations of AlphaFold as a predictive tool, emphasizing that experimental validation - especially thorough biochemical work - remains essential" is not satisfactory.

I don't think major changes are needed to address this concern. Minor figure alterations to present candidate sequences simply labelled by their protein and residue range (not as "closure motif X"), along with adding the models and PAE plots for ecRed1 candidates as mentioned above, should be sufficient. To be clear, I think the authors are being rigorous, and they are clear in the text about what has, and has not, been experimentally validated. My main concern is that a casual reader will draw the wrong conclusions from the figures, for example seeing Figure 4A and Figure 5 and thinking "Ah, ecRed1 has six closure motifs - just like *C. elegans* HTP-3!" when this is emphatically NOT supported by the data.

Specific points:

Figure 4F: Legend appears to be missing, and more importantly it's not very clear what the authors are trying to convey here. To be clear, I understand what they are trying to convey - it's just not obvious from the design of the figure and lack of an accompanying visual aid.

Figure EV8: It is impossible to judge from this figure whether these putative closure motifs in brown algae Red1 proteins are conserved relative to the surrounding (presumably disordered and poorly conserved) protein regions or not.

We thank reviewer 3 for their critical, but constructive comments. We apologise for not having done more to clarify the difference of the predicted and experimentally validated closure motifs. We have undertaken the following steps

- 1) *We now refer to the unvalidated motifs as "Motif" giving rise to M-H2-A, M-R-B etc.*
- 2) *We have added a supplementary figure where we show the structures and PAE plots for each of the predictions for the Red1 motifs (M-R-B to M-R-F)*
- 3) *We have removed the extra motif boxes from Figure 5*

Figure 4F - We have added a figure legend to Figure 4F, and a description within the figure.

Figure EV8 - We would argue that in this figure it is relatively clear that several of these motifs are well conserved, for example CM-H1-A and CM-H1-B are clearly well conserved in the majority of the species compared. We understand that the reviewer's point would be whether, for example, M-R-E is really well conserved within Ectocarpales, or if this is equally poorly conserved as the surrounding region. However, the absence of this motif within other brown algae clades would make it clear, even to a casual reader, that this motif is indeed poorly conserved.

Dr. John Weir
Friedrich Miescher Laboratory
Structural Biochemistry of Meiosis
Max-Planck-Ring 9
Tübingen, BW 72076
Germany

Dear Dr. Weir,

I have now checked all files and am very pleased to accept your manuscript for publication in the next available issue of EMBO reports. Thank you for your contribution to our journal.

Yours sincerely,
